# Replay without sharp wave ripples in a spatial memory task

**John Widloski[1,2] & David J. Foster ●[2] ✉**

Sharp-wave ripples and hippocampal replay are widely viewed as inseparable components of episodic memory consolidation, with ripples broadcasting the episodic content carried by replay sequences. Here, we show that in male rats performing an open-field spatial memory task, replay can occur in the absence of ripples. Replays with and without ripples were organized in virtual space: ripples were confined to discrete "ripple fields," spatially restricted regions defined over the virtual locations depicted during replay and independent of the rat's location. Ripple fields were direction-independent, stable within sessions, and adapted to environmental changes induced by barriers. Ripple fields were matched across animals exposed to the same environment, revealing a conserved spatial code. These results indicate that ripples and replay are distinct but coordinated processes, with ripples selectively tagging a subset of replays linked to learning or novelty. We propose that this coupling enables targeted broadcast of salient experiences for consolidation.

Sharp wave ripples are among the most salient events in the mammalian brain, constituting a transient (~100 ms) high frequency (100–200 Hz) oscillation in the local field potential reflecting a highly coordinated, synchronous firing of neurons across brain regions[1–7] that is well suited for the induction of synaptic plasticity[8–11]. Because ripples occur primarily during rest/sleep and tend to start in the hippocampus, an area long associated with memory[12] and spatial mapping[13], ripples are thought to play a crucial role in episodic memory, by binding complex sensory-motor memory traces to place information in the hippocampus for the purposes of memory recall and consolidation[14–18]. At the same time, hippocampal activity during ripples is highly structured, encoding continuous paths through the environment that evoke the animals own movement through space, albeit at speeds 20 times faster[19–23]. These two phenomena, ripples and replay, are believed to function hand in hand during episodic memory encoding, the former broadcasting the memory contents of the latter, as well as facilitating its consolidation. Supporting this hypothesis, ripples are known to facilitate long-term potentiation and reorganization of hippocampal synapses[24,25] and blocking sharp wave ripples, which tend to co-occur with replay, has been shown to lead to deficits in both memory consolidation and retrieval[26–29].

However, the precise nature of the relationship between ripples and replay remains elusive. For example, replay detectors are generally predicated on the existence of supra-threshold events in the ripple power (ripple) or population spike density (burst)[22,30–35], with ripples and bursts moderately coincident within replay (e.g., 50% of replays in Pfeiffer and Foster[22] contain both; 25–45% from Malvache et al.[36]; 29% in O'Neill et al.[30]). However, it is possible that replays may exist in the absence of ripples or bursts. Indeed, compressed hippocampal sequences in the absence of ripples or bursts are readily observable during run[37–40], though shorter in length than replay, suggesting that ripples are not required for hippocampal sequence generation. In addition, ripples and replay are not always one-to-one: When animals are exposed to long linear tracks or complex environments, replays can be very long, in fact, much longer than the timescale of a single ripple[21,23]. In these cases, multiple ripples tend to occur[21,41–44], suggesting that ripples may be building blocks of replay. Taken together, these considerations suggest that the relationship between replay and ripples may be complex.

To address these questions, we developed a ripple- and burst-independent replay detector and applied it to a large data set consisting of hundreds of simultaneously recorded place cells from rats performing a spatial memory task. Crucially, the task was designed to

[1]Allen Institute for Neural Dynamics, Seattle, WA, USA. [2]Helen Wills Neuroscience Institute and Department of Psychology, University of California, Berkeley, CA, USA. ✉e-mail: davidfoster@berkeley.edu

elicit extended replay in two spatial dimensions, and perturb it with repeated manipulations of spatial and reward contingencies, to generate a wide variety of replay patterns. We find that the hippocampus participates in the encoding of long spatial sequences in the complete absence of ripples. We also show that ripple timing is tightly constrained by the content of replay, occurring in spatially restricted "ripple fields" as a function of the encoded location during replay, and is context dependent, with ripple fields "remapping" across environments when the barriers were moved.

## Results

### Replays can occur in the absence of ripples

To determine whether replay content can exist in the absence of bursts or ripples, we utilized a previously published data set[23] comprising of 6580 replays recorded across 37 session from 3 rats and applied a novel replay detection method that did not depend on the occurrence of population bursts or ripples. While the original dataset included 4 rats, only 3 had sufficient numbers of active place cells to support replay analysis; all sessions from these 3 animals were included in our analysis. Briefly, during each session, rats were trained on a spatial memory task to search for liquid chocolate in a square arena (90 × 90 cm) with movable barriers. Chocolate was available in one of 9 food wells, which alternated on consecutive trials between a learnable fixed location (Home well) and other unpredictable locations (Random wells). Rats learned the location of the Home well after only a few trials, navigating to it quickly even as the barriers changed unpredictably across sessions (Supplementary Fig. 1). We recorded the activity of up to 295 hippocampal place cells simultaneously (mean per session = 182 place cells). Place fields were determined for each active cell by normalizing the spatial distribution of rat positions at spike times during running periods by the rat's position occupancy, and memory-less Bayesian position estimation was used to decode the posterior probability of position from the spiking of all simultaneously recorded cells across overlapping time bins of size 80 ms (shifted by 5 ms) throughout the session. During stopping periods in the task, candidate events were identified as continuous epochs lasting at least 100 ms where the decoded position changed smoothly. Those candidate events that satisfied spatial coverage criteria and passed a place-cell shuffle analysis were classified as replays. Otherwise, no requirements related to sharp-wave ripple power or population spiking were imposed.

Surprisingly, replays possessing long-duration, smoothly varying spatial content were readily detected that were not accompanied by bursts or ripples (Fig. 1A, C). This was true even though replays with strong bursts and ripples were detected within the same session (Fig. 1B, D), even seconds apart within the same stopping period (Supplementary Fig. 2). Next, we defined ripple events as when the ripple power exceeded 2 standard deviations (SD) for at least 15 ms[45], where ripple power was averaged across all tetrodes[22]. Bursts were computed from the population spiking from all well-isolated clusters (see "Methods") as peaks exceeding 3 SD for at least 50 ms[22]. A given replay was considered ripple/burst-less if (1) it did not contain a ripple or burst event (i.e., the time of peak ripple power or spike density was not within the replay) and (2) the peak ripple power and spike density were not greater than chance when compared to random snippets of data throughout the stopping period (see "Methods"). To control for variability in rat behavior during replay, we restricted analysis to replays occurring during reward consumption (see "Methods" for determination of reward consumption times). Within this set, we found that ~24 ± 2% (standard error measure, SEM) of replays did not contain a ripple or burst event.

Considering the fact that averaging ripple power across tetrodes might obscure more "local" ripple events occurring on subsets of tetrodes, we also detected ripples on individual tetrodes. Across replays, the distribution of the number of ripple-detecting tetrodes per replay was U-shaped: ripples tended to be detected by many tetrodes

(peaking around ~48 tetrodes) or by only a few (Fig. 1E, black). The average ripple peak decreased monotonically as fewer tetrodes picked up a ripple (Fig. 1F), demonstrating that there were no large amplitude ripple events occurring on small numbers of tetrodes. At the same time, we saw little relative change in the replay duration (Fig. 1G) as a function of the number of ripple-detecting tetrodes, indicating that the salience of the ripple event is largely independent of replay duration (recall replay duration is determined by constraints on the smoothness of the decoded trajectory). We suspected that the low-amplitude, low-tetrode events could be chance fluctuations in ripple power due to, e.g., locally filtered action potentials[46], which, if occurring independently across tetrodes, would similarly cluster at the lower end of the distribution. Consistent with this hypothesis, repeating the analysis on windows immediately post-replay (+0.75 s) where ripples were unlikely to occur (Supplementary Fig. 3A) yielded only low-tetrode count ripple events (Fig. 1E, green). Thus, while we cannot rule out the possibility of ripple events detected on very few tetrodes, for all intents and purposes we cannot distinguish them from chance. To establish a false positive baseline, for each tetrode, and for each replay, 100 replay-length windows were taken randomly across stopping periods and evaluated for ripples. The probability of ripple detection per tetrode was found to be ~11%. The 95th percentile of the binomial distribution with this success probability across 64 detectors (Supplementary Fig. 3B) occurred at 11 tetrodes, indicating that to detect a ripple event confidently, at least 11 tetrodes must detect it. Note that this estimate is roughly consistent with where the distribution of ripple-detecting tetrodes for the original data intersects with the distribution derived from the post replay periods (~10–15 tetrodes, Fig. 1E).

Incorporating these more sensitive criteria (burst detection, ripple detection on tetrode-averaged ripple power, and ripple detection on at least 11 tetrodes), we found that ~20 ± 2% of replays contained neither ripples or bursts (Fig. 1H). To compare this with chance level, we notice that around 70% of replays have either a ripple or a burst (the sum of the "both" and "ripple only" fractions in Fig. 1H). If we were to assume ripples and bursts occurred independently across replays, then we should expect around 9% of events to be have neither (i.e., prob(either) = (1-prob(ripple)) × (1-prob(burst)) = 0.3 × 0.3 = 0.09). This is far less than the 22% fraction we measured. Using the same logic, we would also expect around 49% of replays to have both a ripple and a burst (prob(both) = prob(ripple) × prob(burst) = 0.7 × 0.7 = 0.49). In contrast, we found that ~80% of replays that contained either a ripple or a burst contained both (Fig. 1H), indicating that ripple and burst events are highly coincident, as expected. This coincidence rate is higher than previous estimates[22,30,36], which we attribute to our large sampling of the dorsal hippocampus. Also, we computed the number of ripple/burst-less replays expected by chance after shuffling place-cell IDs and detecting new replay events (100 times). We found that across sessions, the number of ripple/burst-less replays in the original data was significantly larger than chance (27 out of 37 sessions; $p < 0.001$, Fisher combined probability test; Supplementary Fig. 3C-D). These results suggest that ripple/burst-less replays define a separate, meaningful category of replay.

While ripple/burst-less replays tended to be shorter in duration and occur closer to movement onset than replays with ripples or bursts (Fig. 1I, J), they nonetheless were associated with very low rat speeds (around 1 cm/s on average; Fig. 1K), comparable replay speeds (Fig. 1L), and relatively long average durations (~270 ms, sometimes as long as 1 s -Fig. 1M), ruling out that these events are just theta sequences. Accordingly, close examination of the full local field potential (LFP) power spectrum revealed that while replays containing either ripples or bursts had increased power both in the ripple band (100–220 Hz) and in the sharp-wave band (4–12 Hz) compared to replay-length windows immediately after replay or selected randomly during the stopping period ("random"; Fig. 1N, red vs. green/yellow;

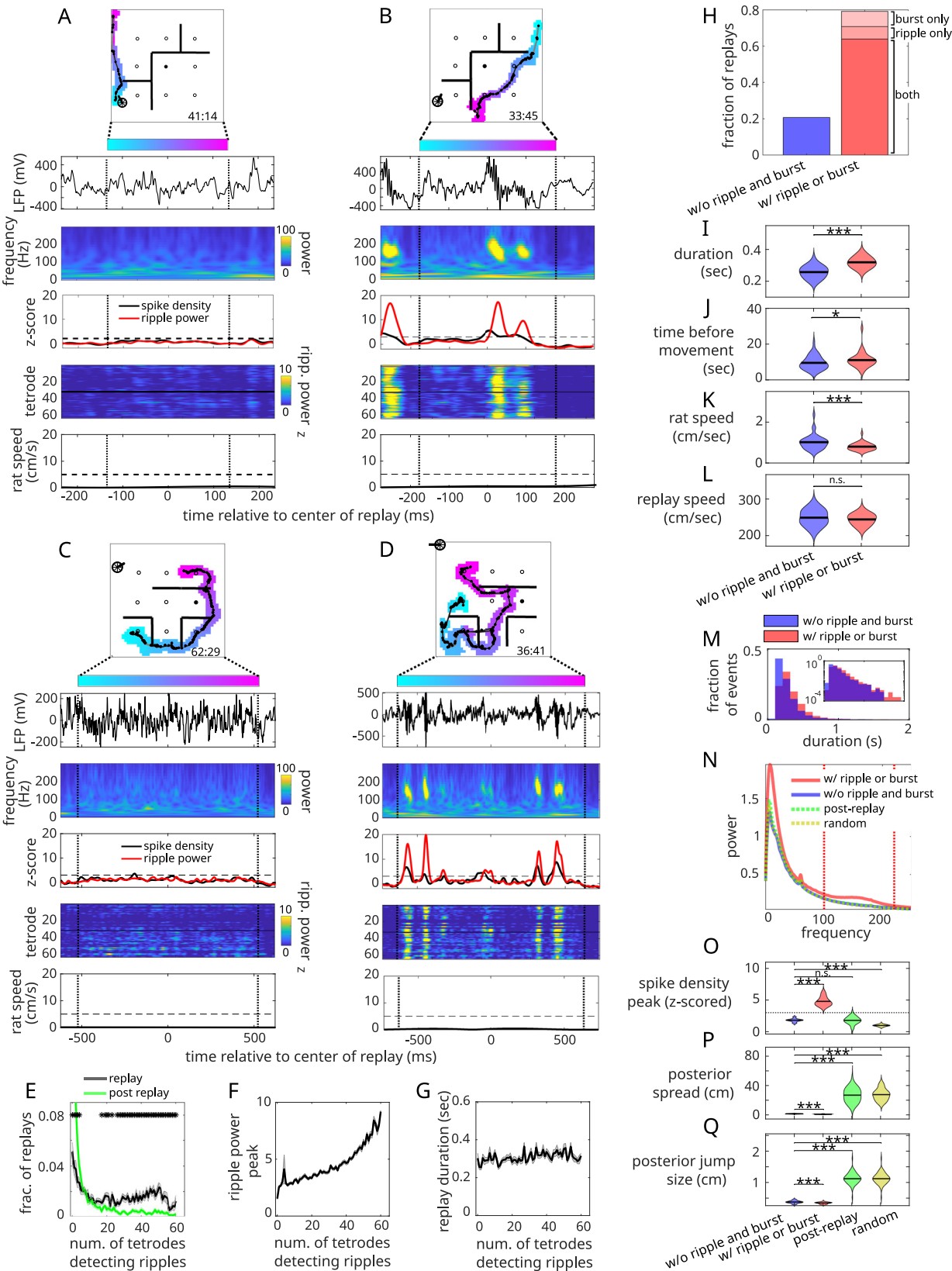

Supplementary Fig. 3E), ripple/burst-less replays did not (Fig. 1O, blue; Supplementary Fig. 3E). Ripple/burst-less replays also showed similar low levels of peak spiking compared to post-replay control periods (Fig. 1O, Supplementary Fig. 3F-G), this notwithstanding the fact that the quality of decoding during ripple/bursts-less replays was substantially improved over replay-length windows taken post-replay or at random (Fig. 1P, Q). This indicates that peak spiking alone provides little information about whether a window contains a replay event. We initially observed a significant difference in decoding quality between replays with vs. without bursts or ripples (Fig. 1P, Q). To account for the possibility that this difference was driven by lower spike rates in ripple/burst-less events, we matched the two replay types on coarse spiking

**Fig. 1 | Replays can occur in the complete absence of ripples and bursts. A** Top row: Example replay from rat 1, session 75. The colored blob is the posterior probability of a replay thresholded at 0.01 and color-coded according to elapsed time within the replay. Solid black line: replay center-of-mass. Time within session is shown at the upper left (min:sec). Black circles and straight lines are the locations of reward and transparent barriers, respectively. Filled circle is the home well. Second row: Raw LFP trace from a single electrode as a function of time within the replay. Black vertical dashed lines demarcate start and end of replay. Third row: Spike density summed across all recorded cells (black) and ripple power averaged across all tetrodes (red). Dashed horizontal line indicates z-score of 3. Fourth row: Ripple power across tetrodes. Tetrodes 1-32 (33-64) targeted the left (right) hemisphere. Fifth row: Rat speed. Dashed horizontal line indicates rat speed of 5 cm/s. **B** Replay with ripples, recorded in the same session as (**A**). Replays without (**C**) and with (**D**) ripples from rat 2, session 87. **E** Fraction of replays in which a ripple was detected by a subset of tetrodes (black) compared to post-replay periods (0.75 s; green), averaged across sessions ($n = 37$ sessions; data presented as mean ± SEM). **F** Ripple power peak and **G** replay duration as a function of the number of ripple-detecting tetrodes, averaged across sessions. **H** Fraction of ripple/burst-less replays (blue), and replays with either bursts only, ripples only, or both (in varying shades of red), averaged across sessions (only means are shown). **I** Replay duration, **J** time of replay before movement, **K** rat speed, and **L** replay speed for ripple/burst-less replays (blue) vs. replays with ripples or bursts (red), averaged across sessions. Black horizontal line indicates median ($n = 37$ sessions; two-sample $t$-tests, two-sided). $P$-values for **I**–**L**: $4.05 \times 10^{-11}$, 0.3, 0.001, 0.3). **M** Distribution of replay durations across sessions. Inset shows log-scale. **N** Power spectral density, and mean **O** peak spike density, **P** posterior spread, and **Q** jump size for ripple/burst-less replays (blue), replays with ripples or bursts (red), post-replay periods (0.75 s; combined across both replay types; green), and random replay-length windows through the stopping period (yellow). Each quantity was averaged across sessions ($n = 37$ sessions; two-sample $t$-tests, two-sided). $P$-values for (**O**), w/o vs. w/ ripple/burst: $4.66 \times 10^{-11}$, w/o ripple/burst vs. post-replay: $1.01 \times 10^{-4}$, w/o ripple/burst vs. random: $3.68 \times 10^{-5}$. $P$-values for **P** (in the same order): $7.3 \times 10^{-15}$, $1.38 \times 10^{-2}$, 0.16. $P$-values for **Q**: $3.74 \times 10^{-13}$, $2.85 \times 10^{-31}$, $1.12 \times 10^{-28}$ (*$p < 0.05$, **$p < 0.01$; ***$p < 0.01$).

properties—specifically, spikes per second and number of active cells per second (Supplementary Fig. 3F-G). This was done by constructing a two-dimensional histogram over both measures for each group and downsampling events so that the resulting histograms were identical. This matching procedure equalized decoding quality across the two replay types (Supplementary Fig. 3H-I), while preserving key differences in ripple power, spike density, and theta power (Supplementary Fig. 3J–L), suggesting that the presence of ripples or population bursts is not required for the generation of high-quality, temporally structured replay events. We also observed ripple/burst-less replays to occur more often at a Random well compared to a Home well just prior to the goal-directed portion of the task (Supplementary Fig. 3M), indicating that these events are more prevalent during the memory-intensive portions of the task. Thus, our method allows us to reveal a large number of spatially extended, high-quality, behaviorally significant replay events that would have gone undetected if classical methods of replay detection had been used.

## Ripples occur reliably as a function of replayed location

To determine whether there was any relationship between ripple timing and replay content, we plotted all replays from a given session with ripple power superimposed (Fig. 2B). We defined the session "ripple field" by summing, for each spatial bin, the instantaneous ripple power for each visit by replay to that bin and dividing by replay occupancy (Fig. 2C). Strikingly, peaks in ripple power did not occur randomly as a function of replayed location but were spatially restricted. Ripple fields were found to exhibit spatial information and stability that was greater than chance for most sessions (35 out of 37 sessions, $p < 0.001$ and 26 out of 37 sessions, $p < 0.001$, respectively; Fig. 2D–F). Chance was determined by recomputing spatial information and stability after circularly permuting the ripple power across all concatenated replays (Supplementary Fig. 4). When the LFP was temporally shifted, spatial information dropped precipitously within tens of milliseconds (Fig. 2G), indicating that the timing between ripples and replay content was tightly controlled. We also ruled out trivial explanations for ripple fields. For example, ripple fields were not artifacts of our decoding approach, since using non-overlapping 20 ms time bins to re-compute replay trajectories yielded similar ripple fields (37 out of 37 sessions, $p < 0.001$; Supplementary Fig. 5B-C). Ripple fields did not reflect locations overrepresented by replay (1 out of 37 sessions, $p = 0.97$; Fig. 2H, I) or the rat's location during replay (0 out of 37 sessions, $p = 0.99$; Fig. 2H, J) Indeed, removing all portions of replay within 30 cm of the rat's location yielded highly similar ripple fields (37 out of 37 session, $p < 0.001$; Supplementary Fig. 5D–F). Lastly, we asked whether ripple fields were directionally selective. Using a density-based clustering algorithm to define field boundaries, we found that for replays that crossed through

each ripple field, ripples occurred independently of the direction of travel (0 out of 22 sessions had directional distributions different from shuffle, $p = 0.99$; Fig. 2K). Taken together, ripple fields bear many of the hallmarks of allocentric coding as seen in single-cell place fields—restricted fields, within-session stability, directional independence—except represented at the population level (via ripples) with respect to imaginary position (replay).

Ripple fields computed separately for each tetrode were highly correlated, even across hemispheres (Fig. 3A, D), suggesting that ripples across the hippocampus are tightly coordinated with respect to spatial content within replay. In contrast, spatial fields derived from the local population spiking on each tetrode were highly variable across tetrodes (Fig. 3B, D), indicating a dissociation between the local LFP and local spiking at each tetrode site. However, spike density fields derived from all clusters across tetrodes (Fig. 3C) were strongly correlated with ripple fields (37 out of 37 sessions, $p < 0.001$; Fig. 3E) and showed similar levels of significant spatial information and stability (36 out of 37 sessions, $p < 0.001$, and 36 out of 37 sessions, $p < 0.001$, significant). These results indicate that ripples and bursts are tightly coupled, that ripples measured locally are representative of the global population dynamics of the hippocampus[47], and that the inhomogeneities of the ripple field are not driven by local biases in place cell sampling.

## Place field over-representation predicts ripple field locations

The evidence for goal-directed behavior (Supplementary Fig. 2) suggests that the rats internalize knowledge of the structure of the environment in order to move flexibly and efficiently through it. This is also supported by the fact that replays conform to the barriers in each session, as described in Widloski and Foster[23]. Thus, we asked whether ripple fields correlated with any aspects of the reconfigured environment or the animal's learned behavior. Strikingly, ripple fields were strongly correlated across rats experiencing the same barrier configuration (7 out of 8 sessions significant, $p < 0.001$; Fig. 4), indicating that ripple field locations were a function of the environmental structure. However, across sessions, ripple fields were only weakly correlated with the proximity to rewards or distance to barriers (5 out of 37 sessions, $p < 0.001$; 6 out of 37 sessions, $p < 0.001$, respectively; Supplementary Fig. 5I-J, N-O), or with changes in the affordances of the environment across sessions, as measured through changes in barrier locations or rat behavior (2 out of 22 sessions, $p = 0.01$ and 2 out of 22 sessions, $p = 0.24$, respectively, Supplementary Fig. 5K-L, P-Q). In contrast, ripple fields were moderately correlated with the rat's trajectory density during movement (15 out of 37 sessions, $p < 0.001$; Supplementary Fig. 5M, R), indicating that ripple fields tended to occupy the corridors of movement through the arena. However, visual inspection of the movement trajectory densities across sessions

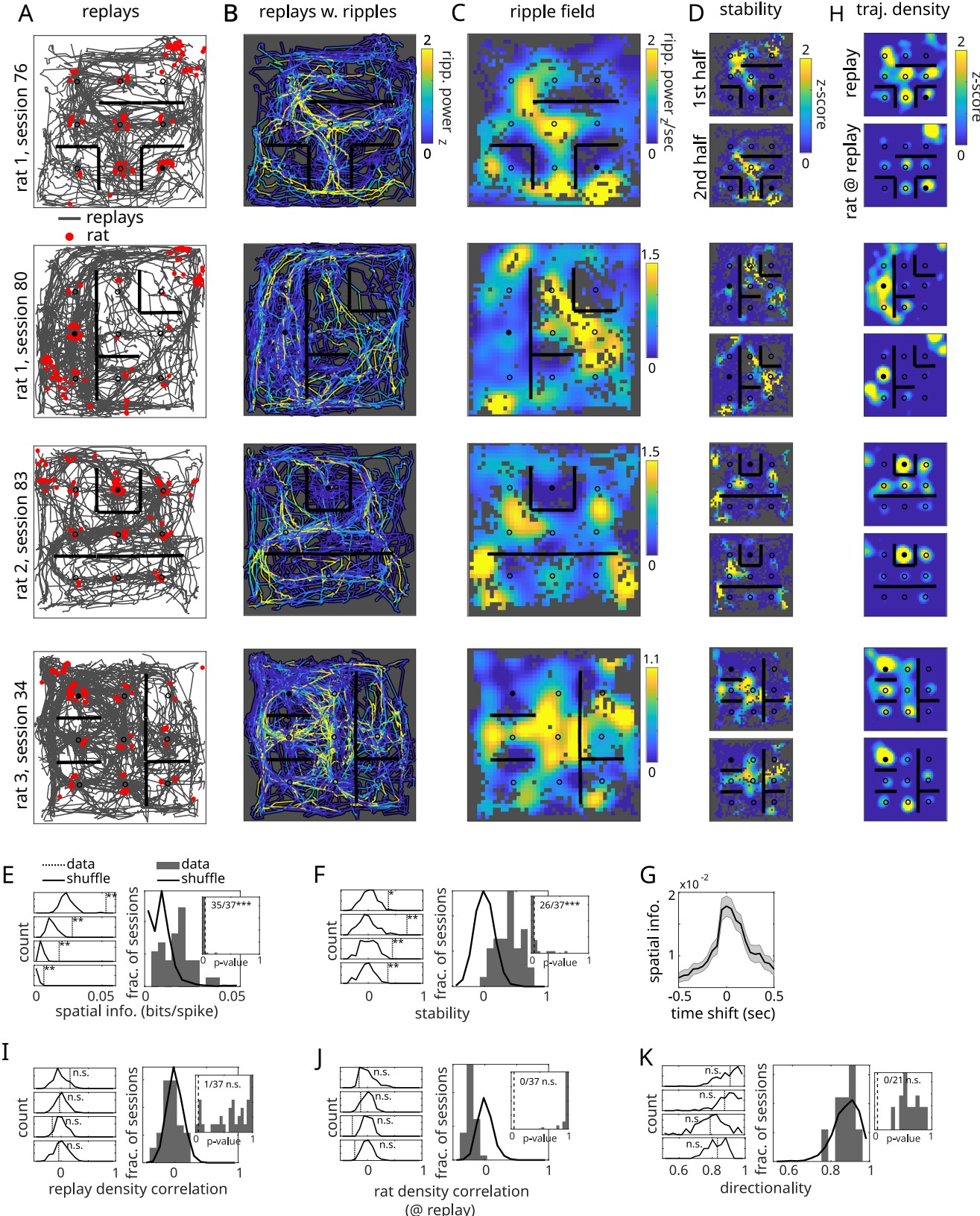

showed that they often failed to account for the idiosyncratic structure of ripple fields (compare Supplementary Figs. 5H to S4M) that was clearly preserved across rats experiencing the same barrier configuration (Fig. 4A), begging the question of whether a stronger correlate of the ripple field locations could be found.

We next examined whether aspects of the hippocampal representation during running predicted the location of ripple fields. Previous work has shown that during replay, individual place cells recapitulate their peak firing rates during run[48–51]. To compare this at the population level, we summed the place fields during run (Fig. 5A)

**Fig. 2 | Ripples occur reliably as a function of encoded location during replay. A** Replays from four example sessions across three rats. Each panel shows all replays recorded within the session (gray traces are replay center-of-masses). Red dots indicate rat location at time of each replay. **B** Replays in (**A**) with z-scored ripple power superimposed. **C** Ripple fields, computed by summing ripple power across spatial bins in (**B**) and normalizing by replay occupancy. Gray bins indicate locations with zero replay occupancy. Ripple fields have units of power/sec. Matrix values have been z-scored across spatial bins for visualization. **D** Ripple fields derived from the first vs. the second half of replays in the session. Values have been z-scored for visualization. **E** Left: Spatial information for the four ripple fields in (**C**) (dashed vertical lines) along with shuffled ripple fields computed by circularly permuting ripple power across replay events (empty histogram). *P*-values computed as the fraction of shuffles greater than the test statistic (one-sided). Right: Spatial information across sessions (filled histogram; *n* = 37 sessions) vs. shuffles (empty histogram; shuffles are combined across sessions). Inset: Distribution of *p*-values across sessions. Dashed vertical line is 0.05. Number of significant sessions

indicated at upper right. Fisher combined probability test used to determine significance of across-sessions *p*-values ($p = 1.55 \times 10^{-31}$). **F** Spatial correlations between split-session ripple fields, plotted as in (**E**) ($p = 3.7 \times 10^{-24}$). **G** Ripple field spatial information as a function of temporally-shifted ripple power, averaged across sessions (data presented as mean ± SEM). **H** Trajectory densities for replay and the rat's position during replay, z-scored for visualization as in (**D**). Spatial correlations between ripple fields and (**I**) replay trajectory density and (**J**) rat position density during replay, plotted as in (**E**) (*p*-values for (**I, J**): 0.97, $2.18 \times 10^{-10}$). **K** Left: Mean vector length of the directional tuning of replays w/ vs. w/o ripples during crossings through ripple field zones (vertical line; see Methods), along with shuffles computed by circularly permuting ripple times across replays (histogram). Right: Directional tuning across ripple field zones (*n* = 22), plotted as in (**D**) along with across-session shuffles. Inset: Distribution of *p*-values for directional tuning across sessions ($p = 0.99$). Dashed vertical line is 0.05. Number of significant field zones indicated at upper right (*$p < 0.05$, **$p < 0.01$, ***$p < 0.001$).

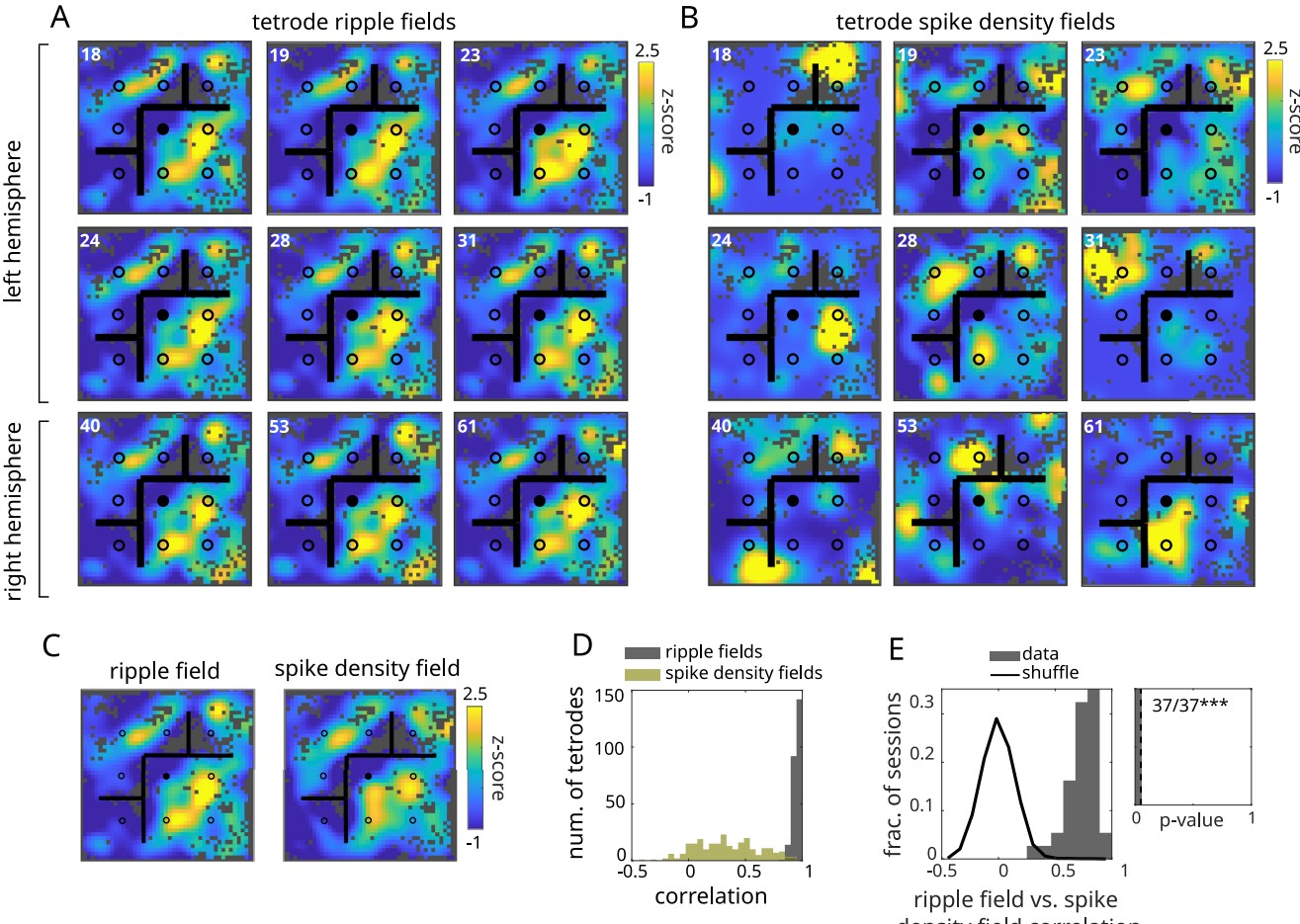

**Fig. 3 | Ripple fields are replicated across tetrodes and hemispheres. A** Ripple fields from a subset of tetrodes from the left (top two rows) and right (bottom row) hemispheres (tetrode number indicated at upper left), z-scored across matrix elements for visualization. Ripple power was computed separately for each tetrode. **B** Same as **A** except using spike density computed locally using only the cells recorded on each tetrode. **C** Full ripple (left) and spike density (right) fields computed by averaging across tetrodes or including all cells recorded across tetrodes, respectively. **D** Distribution of spatial correlations between individual tetrode and full ripple fields (red) and spike density fields (black). Tetrodes were required to have at least 5 well-isolated clusters and mean z-scored ripple power during replay

greater than 0.5 (*n* = 249 tetrodes). **E** Spatial correlations between ripple and spike density fields measured across sessions (solid histogram) along with across-session shuffles computed by circularly permuting ripple power across replay events and measuring spatial correlations between shuffled ripple and unshuffled spike density fields. Inset: Distribution of *p*-values across sessions (*n* = 37 sessions), where *p*-values were computed as the fraction of within-session shuffles greater than the test statistic (one-sided; $p = 2.82 \times 10^{-35}$). Vertical dashed line is 0.05. Number of significant sessions indicated at upper right. Significance computed from Fisher combined probability test (***$p < 0.001$).

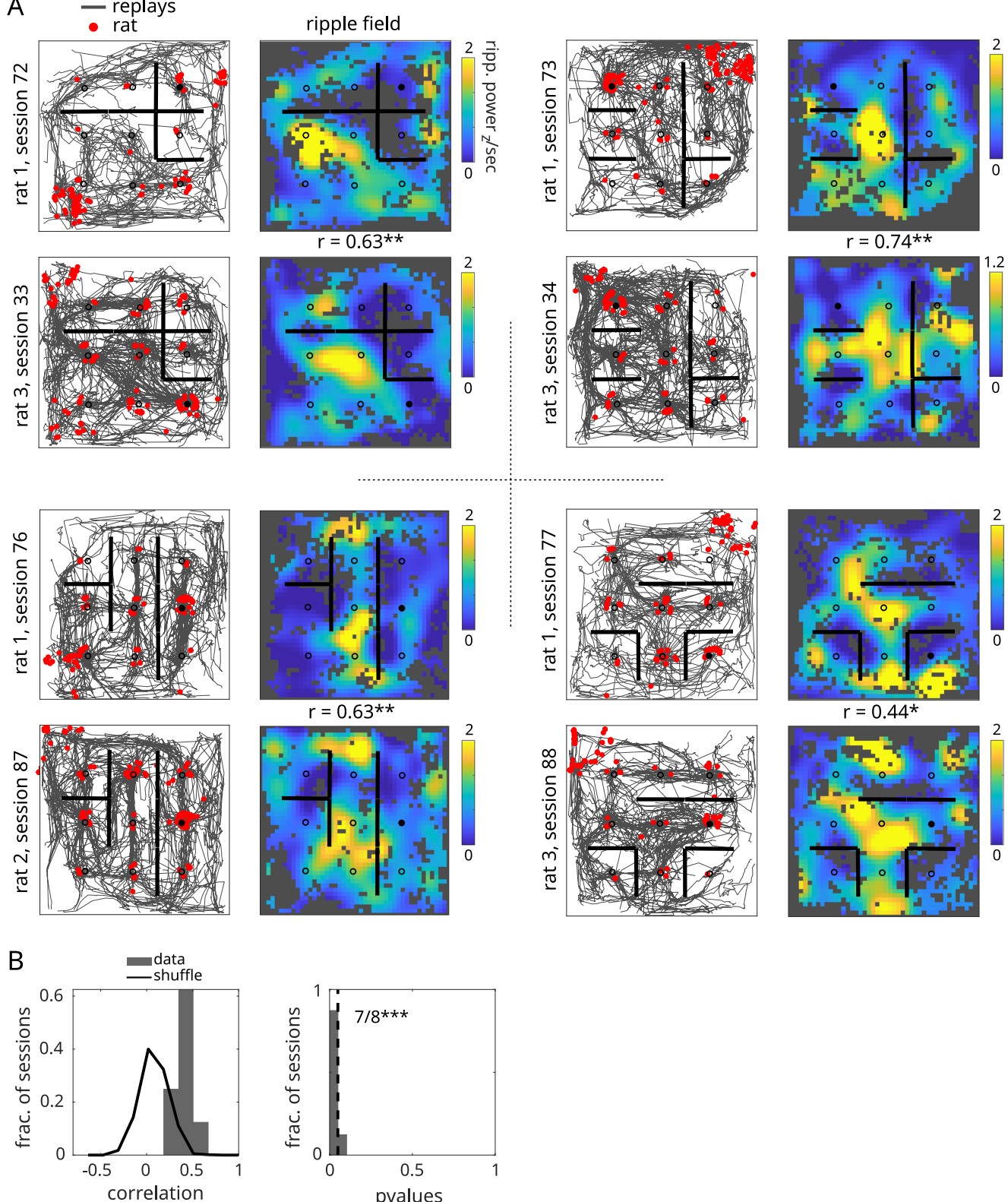

**Fig. 4 | Ripple fields are similar when barrier configurations are matched across rats. A** Replay trajectories (left) and ripple fields (right) for pairs of rats experiencing the same barrier configuration (quadrants). Spatial correlation between ripple fields is reported below. To measure significance, shuffles were computed by circularly permuting ripple power for the second sessions across replay events and measuring spatial correlation. *P*-values were computed as the fraction of shuffles greater than the test statistic (one-sided; $p = 7.14 \times 10^{-8}$).

**B** Spatial correlations between ripple fields across rats for the same barrier configurations ($n = 8$), along with across-session shuffles computed by circularly permuting ripple power across replay events. Inset: Distribution of *p*-values across session pairs, with each session *p*-value computed as in (**A**). Vertical dashed line is 0.05. Number of significant sessions indicated at upper right. Significance computed from Fisher combined probability test (*$p < 0.05$; **$p < 0.01$,***$p < 0.001$).

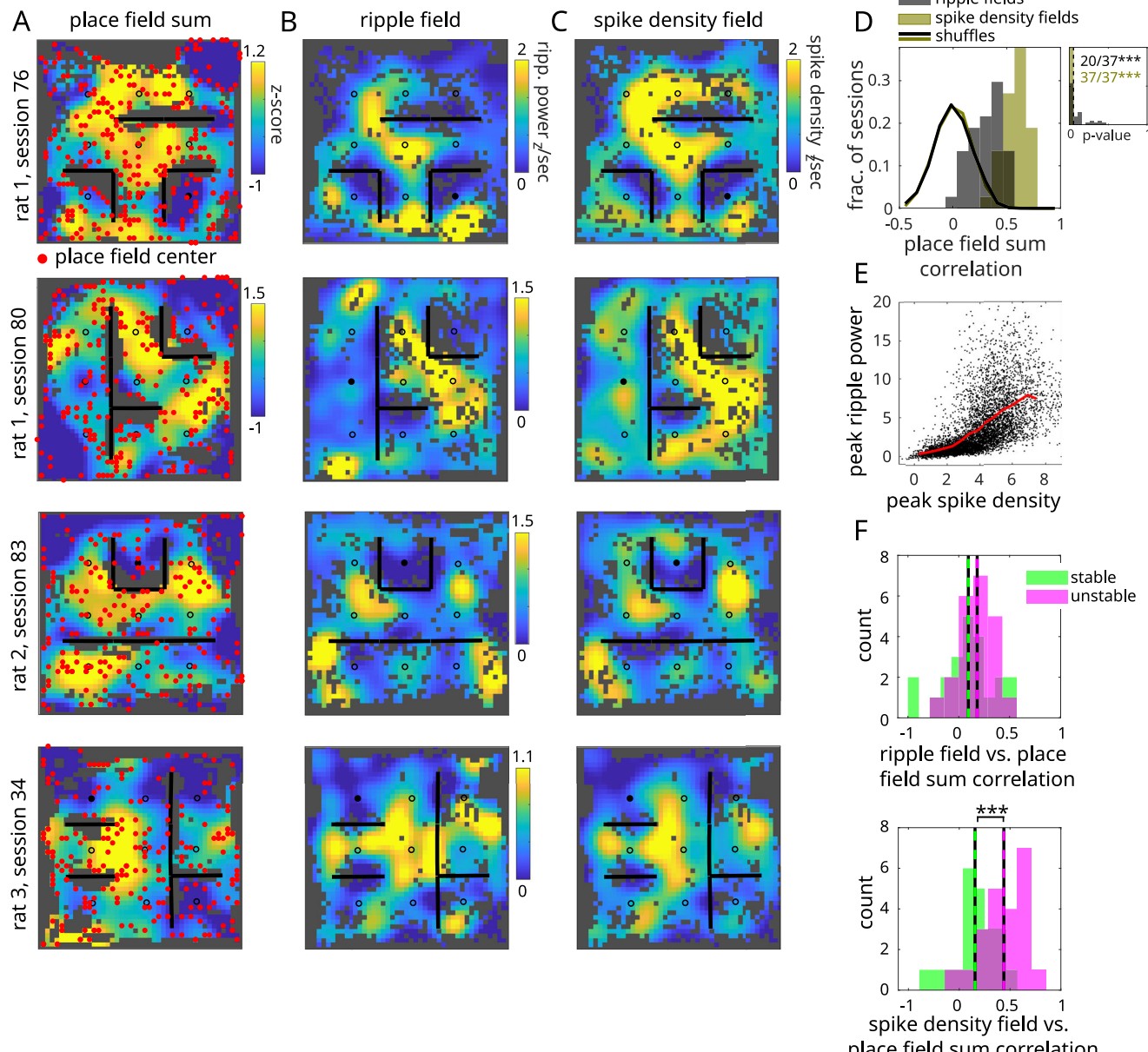

**Fig. 5 | Ripple field locations are predicted by place field over-representation during run. A** Place field rate maps summed across all cells from four sessions across three rats. Matrix values have been z-scored for visualization. Red dots indicate peak field locations across all place cells. **B** Ripple fields and **C** spike density fields from the corresponding sessions in (**A**). Spike density fields have computed by averaging z-scored spike density across spatial bins and normalizing by replay occupancy. **D** Spatial correlations between ripple/spike density fields and summed place field maps across sessions (red/blue histograms, respectively). Shuffles computed by circularly permuting ripple power/spike density across replay events and measuring spatial correlations between shuffled ripple/spike density fields and the summed place field maps (unfilled histograms). Number of significant sessions indicated at upper right. Inset: Distribution of *p*-values across sessions ($n = 37$ sessions), computed as the fraction of within-session shuffles greater than the test statistic (one-sided; $p = 1.05 \times 10^{-17}$). Vertical dashed line is 0.05. **E** Peak ripple power vs. peak spike density across replays ($n = 7273$ replays; black dots), along with mean (red line). **F** Top: Spatial correlations between ripple fields and summed place field maps, for stable (green) and unstable (pink) cells. Stability is measured with respect to the same cell's place field in previous session. Bottom: Same as top but using spike density fields ($n = 22$ sessions; two-sample *t*-test, two-sided, applied to distribution of spatial correlation values for stable and unstable cells measured across sessions; *p*-values: 0.24 for top panel, $8.42 \times 10^{-5}$ for bottom panel; ***$p < 0.001$).

and compared that distribution to spike density fields during replay (Fig. 5B). These distributions were strongly correlated across sessions (37 out of 37 sessions, $p < 0.001$; Fig. 5D), suggesting that place cell recruitment during replay is strongly driven by the density of place fields during run. Likewise, ripple fields were less but still strongly correlated with the summed place fields (20 out of 37 sessions, $p < 0.001$; Fig. 5A, C, D). We hypothesized that the lower correlation in the ripple fields-to-place fields comparison was due to a nonlinearity in the process that converts place cell recruitment during replay into

ripples, as has been suggested[52]. Indeed, peak ripple power and peak population bursting across replay events was strongly nonlinear (Fig. 5E). Together, these results suggest that place cell over-representation during run is a strong determiner of when ripples occur during replay.

Previous work has suggested that ripples preferentially reactivate cells with remapped place fields associated with novel environments or goal locations[53,54]. To probe whether ripple fields were associated with place field reorganization, we divided cells into stable and

unstable groups, depending on whether the cell's place field for the current session was significantly different from the previous session. Summed place field maps for the unstable cells were more strongly correlated with the spike density fields than were the summed maps of the stable cells (Fig. 5F; Supplementary Fig. 6). This is striking given that unstable cells are a minority (30%) in this task (Fig. 5 in Widloski and Foster[23]). In summary, this result shows that ripples selectively echo locations where unstable cells congregate in response to spatial manipulations of the environment.

## Discussion

We developed a replay detection method that did not require detection of ripple or population activity burst and showed that replay can occur without ripples or bursts. To confirm the absence of ripples, we looked for ripples local to one or a few tetrodes, which we were well placed to measure given our 64 tetrodes, a quantity that is vastly more than in previous work. While we found that many ripple events during replay were detected by large numbers of tetrodes, ripples that were detected by small numbers of tetrodes were no more likely than during an equivalent period of time just after each replay, when ripples are unlikely to occur−i.e., these events are below the noise threshold. We also showed that ripple-less replays are not theta sequences, since theta power is particularly low during ripple/burst-less replays. Altogether, we find that a large fraction of replays (approximately one quarter) that would have gone undetected using more traditional replay detection methods.

We found that ripples (and bursts) are bound to the replay of specific information, in the form of certain locations and not others. Previous work has suggested that during exploration of a Y-maze, ripples tended to not occur during reactivation of the maze choice point for replays that crossed from one arm to another, suggesting that ripple timing during replay is not completely random[44]. Our work substantially elaborates on this finding. We were able to define ripple fields, locations encoded by replay where ripples were reliably elicited, and showed that ripple fields exhibited spatial information, within-session stability, and directional independence that was above chance, thus sharing many of the essential allocentric coding properties that belong to single-cell place fields. Crucially, unlike place fields, ripples occurred independently of the rat's location, a remarkable fact given that rat location is such a strong determiner of replay starting locations[19−21]. This implies that ripple occurrences are largely detached from the rat's immediate sensory experience but nevertheless follow an internal logic that is only revealed by recording and decoding large populations of place cells in open environments. Note that our replay detection criteria exclude stationary replays (i.e., non-moving reactivations−Denovellis et al.[31]), which may relate to ripples and bursts according to different organizing principles than the events we have considered here.

Our results challenge the notion that ripples are needed to elicit long replays. The average duration of ripple-less replays was ~270 ms, but we found examples of ripple/burst-less replays that lasted well over 1s (Fig. 1C). We suggest that long, ripple-less replays exist by virtue of the fact that they avoid traveling into ripple zones. On linear tracks, traveling through ripple zones may be unavoidable, which is why replays are nearly always accompanied by ripples (see Fig. 5A in Davidson et al.[21]) and why replay duration is strongly correlated with ripple occurrence[21,43]. While the latter was also true in our data set, this is simply explained by the fact that long replays have, by chance, increased likelihood to cross through multiple ripple zones.

The large prevalence of ripple/burst-less replays, which make up about one-quarter of all replays that we measured, has two further implications. First, it suggests that studies investigating causal manipulations of replay through the detection of ripples[26−29], especially those performed in more open environments, could be missing a great deal of sequential replay-like content and consequently lead to

underestimates of the effects of replay disruption on behavior. Notably, ripple/burst-less replays were more prevalent during memory-intensive phases of the task, further emphasizing their potential functional significance. Second, it challenges the assumption that every offline sequenced reactivation necessarily participates in brain-wide systems-level consolidation. Instead, our results suggest a more hierarchical view of sequence-based retrieval processes operating in the hippocampus, with ripple-less replays being the default mode of offline hippocampal reactivation and used for possibly maintaining the local hippocampal map[55] and its connections to cortex[56], and with replays coincident with ripples participating in the integration of memory traces associated with environmental novelty and change into distributed cortical networks.

An interesting consequence of our finding is that, to the extent that ripple expression is gated by individual spatial fields, the patterns of neural activity driven by ripples may likewise resemble relatively static representations rather than full replayed trajectories. This is consistent with the finding that individual neurons in the prefrontal cortex exhibit activity during hippocampal ripples that is selective for which arm of a Y-maze is being replayed but not for individual locations within the arm[57]. Likewise, VTA neurons have been shown to selectively fire at replayed reward locations during ripples[58], while both the retrosplenial and secondary motor cortices show selective activation to landmarks and cues during reactivations during rest[59,60]. Taken together, these results suggest that what is broadcast from the hippocampus during ripples is a more narrowly restricted set of experiential content than what is expressed in replay, which could explain selective memory for salient experiences[61,62].

Our unique behavioral paradigm using repeated spatial and reward manipulations allowed us to probe the relationship between ripple fields and behavioral context. While ripple field locations were replicated across animals experiencing the same context, they were not tied to reward locations coinciding with rewards or manipulated barriers. Instead, ripple fields were strongly correlated with the density of place fields observed during run, especially locations where the unstable cells congregated in response to the barrier and reward manipulations. This supports the idea that ripples specialize in broadcasting information related to environmental change and is consistent with previous work showing that ripples promote the stabilization of new spatial representations in the hippocampus associated with novel environments, novel reward contingencies, or novel goal locations[14,53,54,63−65]. More broadly, the tight relationship between ripple fields and place field over-representation suggests a new role for the nonlinear reverberatory process thought to underlie ripple generation[52], as reflected in the nonlinear relationship between population spiking and ripple power during replay that we measure. This mechanism may create the spatial distribution of ripple fields as a direct result of where in the environment place fields are over-represented.

Together, our results dissociate two experimental phenomena, ripples and replay, long considered to be aspects of the same process. Moreover, they suggest a novel component of hippocampal-dependent systems memory consolidation, by which certain experiences and not others are selected for further processing. Understanding the functional significance of this relationship will be crucial for elucidating the full spectrum of cognitive processes supported by hippocampal replay.

## Methods

### Experimental model and study participant details

Neural activity was recorded from dorsal hippocampus (region CA1) of 3 male Long-Evans rats (*Rattus norvegicus*; 3−4 months old) performing a goal-directed task in an open field maze with movable barriers (task described below). Rats were housed in a humidity and temperature-controlled facility with a 12-h light-dark cycle. Before the

start of the experiments, rats from the same breeding cohort were housed in pairs. At the start of the experiments, rats were single-housed. All experimental procedures were in accordance with the University of California Berkeley Animal Care and Use Committee and US National Institutes of Health guidelines.

### Task design and training

Rats were trained on a spatial memory task in a square arena to search for liquid chocolate available in one of 9 food wells, which alternated on consecutive trials between a learnable fixed location ("Home" well) and unpredictable other locations ("Random" wells), designated as Home and Random trials, respectively. On each trial a variable time delay (5–15 s) passed before: (A) reward was provided at the bait location, and (B) for all Random trials, a light came on next to the rewarded well, cueing the approach. Before each session, transparent "jail-bar" barriers[66], permeable to visual and olfactory information, were placed in 6 out of 12 possible locations, in a novel, random selection from 924 possible configurations. 2–3 consecutive behavioral sessions were performed per day, each separated by ~3–4 h, and each with a novel barrier configuration (or in some cases, no barriers) as well as a novel, pseudo-randomly chosen Home location (47 sessions total; 12 sessions for rat 1, 17 sessions for rat 2, 8 sessions for rat 3).

### Drive design and surgery

Rats were implanted with microdrive arrays weighing 40–50 g and consisting of 64 independent-adjustable tetrodes made of twisted platinum iridium wires (Neuralynx) gold plated to an impedance of 150–300 MOhms. Drive cannulae were implanted bilaterally to target hippocampal dorsal CA1 (−4.13 AP, 2.68 ML relative to bregma) using a surgery protocol described elsewhere[22,23]. Tetrodes were slowly lowered to the cell layer over the course of 2–4 weeks, which was identified by the presence (and shape) of strong-amplitude sharp-wave ripples. The rats were allowed 3–4 days of recovery, after which behavioral training on the barrier maze task was resumed, but without food restriction in their home cages. Food restriction was resumed a week after surgery.

### Behavioral analysis

Rat position was tracked using automated software from Spike Gadgets and sampled at 30 Hz. Position and speed were smoothed using a Butterworth filter (second order with a cutoff frequency of 0.1 samples/s using the *butter* function in Matlab, selected to give reasonable smoothing to the rat's trajectory). After smoothing, positions were interpolated at 200 Hz (temporal step size of 5 ms) so as to match the sample rate of decoded points within replay (see section on replay detection below). Periods of reward consumption were defined as times in which the smoothed rat speed (a second-order Butterworth filter with a cutoff frequency of 0.02 samples/s applied to the rat's speed computed above, i.e., it was smoothed a second time, dropped below 1 cm/s while the rat was at the rewarded well.

The beginning of each trial was marked as the time at which the rat had moved a distance of 6 cm away from the rewarded well after consuming the chocolate there. Drinking periods were defined as times in which the smoothed rat speed (a second-order Butterworth filter with a cutoff frequency of 0.02 samples/s applied to the rat's speed computed above) dropped below 1 cm/s while the rat was at the rewarded well. During bouts of anticipatory licking, rats exhibited characteristic speed and distance-to-well profiles (see Supplementary Fig. 1 from Widloski & Foster, 2022 for a details). Anticipatory licking periods were thus defined as times in which the rat was both near a well (within 6 cm) and the smoothed velocity stayed within 1–6 cm/s. The parameters listed above for the selection of the drinking and licking bouts as well as the need for secondary smoothing of the rat's speed were determined so as to automate the process of bout demarcation so as to best match what would be selected manually.

To compute the probability of a well visit, a well was counted as visited on each trial if the rat came within 6 cm of it at least once. Well visit probability was then calculated in two ways, as a function of trial and also collapsed across trials, respectively. For the former, well visit probability as a function of trial number was defined as the total number of times a particular well (Home vs. Random) was visited on the trial divided by the total number of sessions. For the latter, well visit probability was defined as the total number of times a particular well was visited across trials divided by the total number of trials, then averaged across sessions. For determining well visits, only well visits that occurred within 5 s of the start of the trial and at least 1 s before the start of the drinking period were counted. The latter constraint was imposed so as to ensure that the behavior analyzed was unaffected by possible reward cues. Only trials with duration less than 60 s were considered. For both the Random-well visit probabilities and Random-well anticipatory licking durations, data was averaged across all 8 Random wells of the session. For the Home well shuffle, the Home well was selected at random 10 times and the well visit probabilities were recomputed and averaged.

### Cluster analysis

Spikes were extracted from channel LFPs, sampled at 30 kHz and referenced against a tetrode placed in the corpus callosum (one for each hemisphere), using Spike Gadgets Trodes software and clustered automatically using Mountainsort[67] and merged across sessions using the *msdrift* package. Additional cluster mergings across sessions was performed manually based on similarity of waveform. Clusters were accepted if noise overlap <0.03, isolation >0.95, peak SNR >1.5[67] and had passed a visual inspection.

### Place fields

In order to compute the cell's place field, for each spike that occurred during run (rat speed >10 cm/s), the rat position was found through linear interpolation (*interp1* in Matlab). Positions were binned with 2 cm square bins. The unsmoothed rate map for the $i$th cell was defined as

$$\tilde{f}_i(\mathbf{x}_j) = \frac{\text{\# of spikes fired within the } j^{th} \text{ spatial bin centered at } \mathbf{x}_j}{\text{time spent within the } j^{th} \text{ spatial bin centered at } \mathbf{x}_j}.$$

Smoothed rate maps, denoted as $f_i(\mathbf{x}_j)$, were computed by convolving the rate maps with a 2D isotropic Gaussian kernel (8 cm standard deviation (SD), 90 × 90 cm kernel size) using the *nanconv* function in Matlab with arguments "edge" and "nanout", which corrects for boundary effects and unvisited bins.

Spatial information (bits/spike) for the $i$th cell was defined as

$$SI_i = \sum_{j=1}^{L} \tilde{P}(\mathbf{x}_j)\left(\frac{f_i(\mathbf{x}_j)}{r_i}\right)\log_2\left(\frac{f_i(\mathbf{x}_j)}{r_i}\right)$$

where $L$ is the number of spatial bins, $\tilde{P}(\mathbf{x}_j)$ is the probability of the rat or replay being at the $j$th spatial bin, and $r_i = \sum_{j=1}^{L} \tilde{P}(\mathbf{x}_j)f_i(\mathbf{x}_j)$ is the cell's mean firing rate. Place cells were identified as having place fields with $r > 0.01$ Hz and $SI > 0.5$ bits/spike.

To determine place field stability, first rate map correlations were defined as the Pearson's correlation between any pair of rate maps (i.e., place fields). For the $i$th cell, with rate maps $f_i^I$ and $f_i^J$ for the $I$th and $J$th sessions, respectively, the rate map correlation was

$$\rho_i^{RM} = \frac{\sum_{j=1}^{L}\left(f_i^I(\mathbf{x}_j) - \langle f_i^I \rangle\right)\left(f_i^J(\mathbf{x}_j) - \langle f_i^J \rangle\right)}{\sqrt{\sum_{j=1}^{L}\left(f_i^I(\mathbf{x}_j) - \langle f_i^I \rangle\right)^2 \sum_{j=1}^{L}\left(f_i^J(\mathbf{x}_j) - \langle f_i^J \rangle\right)^2}},$$

where $\langle f_i \rangle = \frac{1}{L}\sum_{j=1}^{L} f_i(\mathbf{x}_j)$ is the mean spatial firing rate. Rate map correlations were evaluated only at visited spatial bins common to

both sessions and were only measured for cells identified as place cells for both sessions and whose rate maps had minimal barrier overlap in both sessions. Barrier overlap was assessed as follows: First, spatial bins were denoted as "active" if the firing rate density in that bin was greater than 1 Hz/cm. Rate maps for which at least 60% of all active bins were at least 4 cm away from the nearest barrier were considered to minimally overlap with the barriers. A rate map correlation shuffle distribution was computed for each place cell by randomly permuting place cell ID's 100 times in the second session and recomputing the correlations. A place cell was called stable across a pair of sessions if its rate map correlation exceeded the 95th percentile of its shuffle distribution; otherwise, it was called unstable. Field centers were calculated using a density-based clustering approach. First, rate maps were treated as discrete probability distributions and resampled 2500 times (using the *pinky* function in Matlab). Then, the sample points were clustered using *dbscan* in Matlab, with a neighborhood search radius of 2.5 bins and a minimum number of neighbors of 50. Field centers were calculated as the center-of-mass (COM) of all points belonging to the same cluster.

## Bayesian decoding

Let $k_i$ be the number of spikes emitted by the $i$th place cell in a given time bin of duration $\tau$. For all analysis, a time bin duration of 80 ms was used. The posterior probability at bin $\mathbf{x}_j$ conditioned on the activity vector $\vec{k}$ (with the $i$th element as $k_i$) is given by Bayes rule (assuming Poisson spiking noise statistics, independence between neurons, and a uniform spatial prior[21]):

$$P(\mathbf{x}_j | \vec{k}) = \prod_{i=1}^{M} P(\mathbf{x}_j | k_i) \propto \prod_{i=1}^{M} f_i(\mathbf{x}_j)^{k_i} e^{-\tau f_i(\mathbf{x}_j)},$$

where $M$ is the number of neurons. A uniform prior was used for the purposes of making minimal assumptions about the location of the decoded positions. The posterior probability was computed for all bin locations $\mathbf{x}_j$ where $1 \le j \le L$ and $L$ is the total number of spatial bins. Define $P_j = P(\mathbf{x}_j | \vec{k})$ and let $\mathbf{x}_j = (x_j, y_j)$ be the components of the $j$th spatial bin. The components of the posterior COM were given by

$$\mathbf{x}_{cm} = [x_{cm}, y_{cm}] = \left[ \sum_{j=1}^{L} x_j P_j, \sum_{j=1}^{L} y_j P_j \right].$$

The posterior spread was defined as the square root of the second central image moment of the posterior:

$$m^2 = \sum_{j=1}^{L} \left( x_j - x_{cm} \right)^2 \left( y_j - y_{cm} \right)^2 P_j.$$

The posterior COM jump size was defined as the L2 norm of the difference vector between consecutive posterior COM estimates:

$$\delta = \| \mathbf{x}_{cm}^t - \mathbf{x}_{cm}^{t+1} \|.$$

## Replay detection

We briefly outline the procedure for detecting replays, which has been previously published[23]. The Bayesian decoder was applied to spikes within a sliding window of 80 ms duration (shifted in 5 ms increments) over the entire session from all place cells found in the session. Time bins were kept for further analysis based on three criteria: rat speed ($v_{rat} < 5$ cm/s; rat speed was computed at the center of each time bin via linear interpolation), posterior spread ($m < 10$ cm), and posterior COM jumps size ($\delta < 20$ cm). We defined a candidate replay as a set of temporally contiguous bins satisfying the above criteria. Sub-sequences captured epochs in which the posterior was well defined (small posterior spread) and moved smoothly (small COM jump size

across time steps). Neighboring sequences were merged if the spatial and temporal gap between them was 20 cm and 50 ms, respectively.

A candidate replay (merged or not) was denoted a replay if: (1) its duration was greater than 100 ms, and (2) it's spatial dispersion $\mathcal{D}$ was greater than 12 cm, where spatial dispersion was defined as

$$\mathcal{D}^2 = \frac{1}{M} \sum_{t=1}^{M} \| \mathbf{x}_{cm}^t - \langle \mathbf{x}_{cm} \rangle \|,$$

with $M$ is the length of the sequence and $\langle \mathbf{x}_{cm} \rangle = \sum_{t=1}^{M} \mathbf{x}_{cm}^t$, and (3) it passed a place cell-ID shuffle test: Each event was re-decoded using shuffled place-cell IDs 100 times, and the mean posterior spread and jump size was computed for each event. $P$-values for each replay were computed as the fraction of shuffle events less than the actual value for both measures. Replays were required to have $p$-values for each measure less than 0.05.

## Ripple and burst event detection

Sharp wave-ripple amplitude (denoted as "ripple power") was computed for each tetrode by band-pass filtering the LFP on one its four channels in the 100 to 220 Hz range and extracting the magnitude of the Hilbert transform. Population spike density was computed by first summing the total number of spikes from all clusters within a session (i.e., clusters with noise overlap < 0.03, isolation > 0.95, peak SNR > 1.5) in 1 ms non-overlapping time bins. Both the ripple amplitude and spike density were smoothed through convolution with a Gaussian kernel (80 ms SD, 1000 ms kernel size) and z-scored, unless when ripple power was averaged across tetrodes, in which case z-scoring occurred after averaging. The mean and standard deviations used for z-scoring were computed from stopping periods only (i.e., rat speed <5 cm/s). Peak events in stopping period data were visually inspected to make sure that z-scoring was not biased by "noise" events (e.g., chewing artifacts, implant collisions, scratching, etc.). The continuous wavelet transform (cwt function in matlab) was used to compute the LFP power spectrogram over time (e.g., Fig. 1A), where power was computed as the magnitude of the wavelet coefficients. The fast Fourier transform (*fft* function in Matlab) was used to compute the LFP power spectrum (e.g., Fig. 1K), where the power was given as the magnitude of the Fourier transform coefficients.

Candidate ripple events were defined as when the z-scored ripple power (on individual tetrodes or averaged across tetrodes, depending on the analysis) peaked above 2 standard deviations and lasted for at least 15 ms. Event start and end times were defined as when the ripple power returned to the mean. Adjacent ripple events were merged if the time boundaries were less than 50 ms apart. A given replay was determined to be ripple-less if it (1) contained no ripple events according to the ripple detection procedure (by "contain", we mean that the peak ripple power time was not within the replay) and (2) passed a ripple power shuffle test requiring that the peak ripple power within the replay not exceed the 95th percentile of a shuffle distribution comprised of ripple power peaks taken from 100 random equal-length snippets of LFP across stopping periods but outside of other replay times.

Candidate burst events were defined as when the z-scored population spike density peaked above 3 standard deviations and lasted for at least 50 ms[22]. Event start and end times were defined as when the spike density returned to the mean. A given replay was determined to be burst-less if it (1) contained no burst event (i.e., peak burst time was not within the replay) and (2) pass a spike density shuffle test requiring that the peak spike density within the replay be within the 95th percentile of the shuffle distribution comprised of spike density taken from 100 random equal-length snippets across stopping periods but outside other replay times.

## Ripple and spike density fields

To compute ripple and spike density fields, both tetrode-averaged ripple power and spike density were interpolated to replay time scale (20 Hz). Ripple fields (spike density fields) were then computed by summing ripple power (spike density) across all replay visits to a given spatial bin (2 cm) and normalized by replay occupancy. Smoothed ripple and spike density fields maps were computed through convolution with a 2D isotropic Gaussian kernel (8 cm standard deviation (SD), 90 × 90 cm kernel size) using the *nanconv* function in Matlab (with arguments "edge" and "nanout"). Spatial information was computed as for place fields described above, replacing the rate map with the ripple field or spike density field, and the rat occupancy with replay occupancy. Ripple field zones were calculated using the same density-based clustering approach used to find place field centers. First, ripple fields were treated as discrete probability distributions and resampled 10,000 times. Then, the sample points were clustered using *dbscan* in Matlab, with a neighborhood search radius of 3 bins and a minimum number of neighbors of 250. Only clusters with at least 1000 samples were considered. For each of these clusters, the boundary was taken as the level set of the cluster density equal to 0.3 of the peak value. After defining the ripple zone boundary, replay passes through the boundary were extracted. Only passes with a minimum length of 30 cm were considered. The direction associated with each replay pass was computed as the average angle of all segments of the pass (replays were sampled at intervals of 5 ms as described above).

## Barrier dissimilarity

Local barrier similarity is a measure of the local environmental structure across two barrier configurations, as defined in Widloski and Foster[23]. First, the barrier potential was computed for each barrier configuration by convolving the barriers with a 2D isotropic Gaussian kernel (10 cm SD, 90 × 90 cm kernel size). Let $\{\mathbf{x}_k^b\}$ be the set of bin overlapping with the barriers. The barrier potential at the $i$th spatial bin was computed as

$$b(\mathbf{x}_i) = \sum_{j=1}^{L} \sum_{k=1}^{K} \delta(\mathbf{x}_i - \mathbf{x}_j - \mathbf{x}_k^b) h(\mathbf{x}_j).$$

where $h$ is the Gaussian kernel and $K$ is the number of barrier-overlapping spatial bins. We defined the barrier dissimilarity ($BD$) at the $i$th spatial bin across a pair of sessions $I$ and $J$ as

$$BD(\mathbf{x}_i) = \frac{\sqrt{\sum_{l=1}^{4} \left( b_l^I(\mathbf{x}_i) - b_l^J(\mathbf{x}_i) \right)^2}}{||2h||} - 1,$$

where the summation is over the 4 closest spatial bins to the $i^{\text{th}}$ bin and $||.||$ is the L2 norm.

## Rat trajectory dissimilarity

Like the barrier dissimilarity across sessions, the rat trajectory dissimilarity measures the local dissimilarity of the rats trajectory across two barrier configurations. First, for each session and for each spatial bin, the distribution of angles for all segments of the rat's trajectory that exist inside a circle of radius 8 cm centered around that bin were computed. The rat trajectory dissimilarity across a pair of sessions was computed as the negative correlation between the two angle distributions at each spatial bin.

## Reporting summary

Further information on research design is available in the Nature Portfolio Reporting Summary linked to this article.

## Data availability

The processed data generated in this study have been deposited in the Zenodo database under the accession code https://doi.org/10.5281/zenodo.16916108.

## Code availability

All custom-written MATLAB code has been deposited in the Zenodo database under the accession code https://doi.org/10.5281/zenodo.15199609.

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

## Acknowledgements

We would like to thank Matt Kleinman, Caitlin Mallory, and David Theurel for helpful discussions. This work was funded through NIH grant NS113557.

## Author contributions

J.W. conceived of and designed the study, acquired the data, and performed the analysis. J.W. and D.J.F. wrote the manuscript.

## Competing interests

The authors declare no competing interests.
