## [Transparent Peer Review file · Nature Communications]

Replay without sharp wave ripples in a spatial memory task.

Corresponding Author: Dr John Widloski

Version 0:

Reviewer comments:

Reviewer #1

(Remarks to the Author)

Summary

The title of this work refers to an observation of a somewhat-weakly supported phenomenon ("Replay without sharp wave ripples"). However, the subsequent analysis of the structure of population bursts / ripple oscillations during sequences of spiking produced when animals are paused during a continuous exploration task is very exciting and potentially of high significance. Taken together, this work undoubtedly merits being reported in this platform.

Sequences generated in the hippocampus have been conceptually identified as potentially critical features for learning and memory - facilitating long term storage, supporting recall, or even simulating potential future behaviors based on a model of the environment. However, a significant portion of the evidence for these ideas relies on a bit of slight-of-hand - the authors talk about sequential activation, but then focus on sharp wave ripples. Sharp wave ripples are a high-frequency oscillation associated with highly synchronous bursts of activity from ensembles of hippocampal neurons (population burst events, PBEs). Hippocampal replay during slow-wave sleep and awake quiescence has long been asserted to co-occur with SWR/PBEs. (SWR are defined based on local field potentials and PBEs are defined based on population firing.)

While previous authors have pointed out that not all SWR occur during a PBE and vice versa, and that hippocampal sequences can continue across chains of SWR, this work for the first time focuses on analyzing hippocampal activity during stopping periods based on decoded trajectories, and finds that many stopping periods are associated with continuous decoded trajectories which may or may not contain SWR. Moreover and very excitingly, they find that the SWR occur during decoded trajectories in consistent spatial locations associated with spatial learning. Again, while previous work had identified that SWR-linked replay contains a non-uniform distributions of represented spatial locations, this was an isolated observation that did not recognize that these ripples might be occurring as segments in part of longer events. Thus, this work will focus further investigations into the distinction between replay and SWR as complimentary but potentially functionally and mechanistically distinct phenomena.

While my enthusiasm for the work is high, the manuscript could be improved by clarify a number of methodological uncertainties.

Major Concerns:

(A) The overarching claim that 25% of replays contained neither ripples nor bursts rests on the definition of (1) replays, (2) ripples, and (3) bursts.

(A1) Replays are defined as periods when the Bayesian posterior (from place cells) meet 3 criteria - speed < 5 cm/s, posterior spread < 10 cm, posterior COM jumps < 20 cm. While the arena size is not given, previous work suggests it is ~90 cm on a side. Thus the example event in Figure 1A comprises a trajectory of 90 cm over 500 ms, or a speed of 180 cm/s (which is >> 5!). Presumably, this implies that the speed threshold is a typo of some sort.

(A1b) Replay segments are combined if the gap between them is < 20 cm and < 50 ms. But no minimum size of a segment is described. So this seems to suggest that as the 5 ms sliding window decodes slide along, two could connect 50 ms apart and create a replay segment even if the intervening 40 ms of bins did not satisfy the other criteria. Since the minimum duration of a replay segment is 100 ms, this suggests that an event could be constructed by three 80 ms windows, each 40 ms apart, which happened to have nearby decoded positions. ****Critically**** these special moments could largely be during ripples, and the smooth trajectories between them could just be the interpolation generated by sliding windows. It is unclear

how to increase confidence here, but it would be very reassuring if non-overlapping bins generated consistent results?

(A1c) Unlike classic replay analysis, no effort appears to be made here to identify portions of decoded trajectories that are "significant". While that is sensible, the big concern is that non-SWR replay is being detected effectively by chance - spikes from a small number of neurons are able to drive the appearance of a sequence. Rather than asking for shuffle analyses, is it possible to justify NOT doing them? At minimum, it would be worth reporting how the replay criteria (speed, jump distance, posterior spread) compare with shuffled data.

(A2) Is the z-scoring of ripple power done only during periods when the animal is stopped, or the whole session? Does it also include sleep epochs not discussed in this paper? What about periods of noise from, e.g., scratching or chewing artifacts?

(A3) Population burst events are described as drawing from "all clusters". Does this include multiunit activity? Clear artifact/noise? Obvious interneurons? Are the results roughly equivalent if PBEs are defined just based on pyramidal cells?

(A4) "... approximately 25% of replays contained neither ripples or bursts". It would be clearer to say that "~25% of replay segments shared no overlap with ripple or population burst event periods." ("Contained" might alternatively mean "completely included" rather than "overlapped with".)

(B) The definition of a ripple occurrence using the average ripple power across all tetrodes is probably not consistent with the literature. The "bimodal" distribution in Figure 1E has a peak at 50 tetrodes, which is vastly more than most historic recording experiments. This needs to be acknowledged in the discussion. (It also is unclear that it is bimodal?)

Moreover, the idea that there are "chance" ripples is a semantic sleight-of-hand. It would be much fairer just to structure this paragraph around the idea that many SWR occur as small events on a limited number of tetrodes, but that "large ripples" or "ensemble ripples" represent a unique class with special properties.

(C) I could not understand Figure 1H (are the lighter salmon colors bars that are overlapping the darker "both" bar or separate, taller bars?), and the discussion was also quite confusing. I think the upshot is that SWR and PBEs largely measure the same phenomenon. This is not surprising, but perhaps since the Malvache, 2016 reference is in the literature, it is worth including. (Also relevant for cross-hemisphere correlation, which you could report as a useful number!) But I think that reporting it as unexpected or unlikely makes it hard on a reader who is used to the two being considered the same events.

(D) It is important to report the way that LFP signals are electrically referenced, and how electrode positions are inferred particularly when discussing averaging/comparing ripple signals across different tetrodes. Is it valid to average ripple band power values from tetrodes in s. pyramidae and s. radiatum, for instance???

(E) How do ripple fields correspond with spatial occupancy?

Minor Concerns:

(1) The arena size is not specified in the text or in the figures (i.e., as a scale).

(2) The length of the filter for the Gaussian smoothing kernels should be reported, not just the SD.

(3) Can the rats hear the next reward well being filled? Does this occur during exploration or when they are still? Is it associated with the location of the ripple fields? What about their average location when the reward light is illuminated?

(4) The discussion should probably discuss "ripples that aren't replay", and how they fit into the story. For example events when the posterior is still or loops back on itself.

(5) "We restricted analysis to all replays associated with reward consumption" - this could be more clearly phrased based on a definition.

(6) "To establish a false positive baseline, we performed the same ripple-detection analysis on post-replay windows (0.75 sec after the end of the replay), where we determined ripples to occur at a minimum (Figure S2A)." Both black and green lines speak of special neural states. It is a bit confusing using the post-replay window as the false positive baseline, especially when this window is selected by finding the minimum ripple detection. Why not randomly select windows of the same length of replays (like Figure S2B)?

(7) Page 13, "An interesting consequence of our finding is that, to the extent that ripple expression is gated by individual spatial fields, the patterns of neural activity driven by ripples may likewise resemble relatively static representations rather than sequential trajectories." What is the corresponding result that supports this statement?

(8) (Methods) What is the unit for spatial information? It is said to be in bit/spike but the SI criteria for identifying a place cell is in bit/second.

(9) Replay detection without SWR/PBE event detection first has been used before, e.g., Carey, Tanaka, van der Meer 2019 has a "sequence detector". Should be cited/mentioned?

(10) Figure 1G needs a unit (seconds?).

(11) Figure 5 B/C captions don't match labels in figure.

Reviewer #2

(Remarks to the Author)

Reviewer #3

(Remarks to the Author)

Widloski and Foster's study provides evidence for the existence of ripple-less replays, which they describe as a functionally distinct subset of hippocampal activity. These replays were characterized by smooth, continuous spatial trajectories yet lacked the sharp-wave ripples traditionally associated with hippocampal memory processes. While their temporal properties suggest a potential role in hippocampal function, the claim of functional distinctiveness is not directly tested, leaving room for alternative interpretations such as their alignment with other hippocampal phenomena like theta sequences. Furthermore, the concept of "ripple fields," spatially stable zones where ripple-associated replays occur, is introduced as a mechanism for amplifying salient experiences during memory consolidation. However, the connection between ripple fields and learning or memory remains hypothetical, as no direct link to improved task performance or learning outcomes is provided. Without metrics such as accuracy or learning curves, it is unclear whether the three rats genuinely learned the task or relied on simpler, non-mnemonic strategies.

Furthermore, the study lacks a careful theoretical and experimental decomposition of the behavior, which is crucial for understanding how replay without sharp-wave ripples relates to functional outcomes.

Version 1:

Reviewer comments:

Reviewer #1

(Remarks to the Author)

We remain very enthusiastic about this paper. The remarkable data set that the authors have gathered permits them to study sequential non-theta patterns in hippocampus in a way that provides much better grounding than we have had previously. In addition, the "ripple field" phenomena seems potentially important and will merit further study by the field. Most importantly, clearly articulating the independence of sharp wave ripples and sequence generation will be of critical relevance to our understanding of how the system works for memory consolidation and integration.

That said, we did have some residual concerns in the most recent draft we have received.

(1) In the textual introduction to replay detection, "114-115 decode the posterior probability of position from the spiking of all simultaneously recorded cells across time bins of size 80 msec throughout the session. During stopping periods in the task, candidate events were identified as continuous epochs lasting at least 100 msec" - This (80 vs 100) will confuse the reader and force them to discover the fact that a sliding window is employed. Consider adding this information here?

(2) Now that replay detection is clarified to use a shuffle test, Figure 1P,Q/S2I,J and the text callouts (lines 185-190) are problematic. First, the replay events are selected by shuffle testing the posterior spread and jump distance to be small. It thus seems unsurprising (and almost a tautology) that they would be smaller than randomly chosen data (green and yellow) or particularly compared to shuffled data (S2I,J)? Second, in particular, the claim "the quality of decoding during ripple/burstless replays vs. replays with bursts or ripples was comparable" seems unsupported by the data that shows a highly statistically significant difference between them. This claim is paired against the first claim ("they are also much smaller than random"), but as pointed out, that's obvious by construction. Perhaps the idea that you hope to demonstrate to the reader - that these are also real replay - could be made by making a statement about how much the distributions overlap? ("the median ripple/burtless replay had a quality greater than 40% of replays with bursts or ripples").

Reviewer #2

(Remarks to the Author)

Reviewer #3

(Remarks to the Author)

Here, I restate my core concerns in a more comprehensive and focused manner. I hope this clarifies my original intent, and I apologize if these concerns were not expressed clearly enough in the initial round of review.

The main claim of the manuscript is that hippocampal replay can occur in the absence of sharp wave ripples and that ripple occurrence is spatially organized into stable “ripple fields” in decoded space. The authors propose that ripples selectively tag behaviorally relevant replays, particularly those associated with learning or novelty, challenging the conventional view that replay and ripples are functionally inseparable.

The manuscript presents compelling electrophysiological evidence in support of this novel framework. However, the interpretation that ripple-less replays reflect memory-related processes hinges critically on the assumption that animals were engaged in a spatial memory task. The problem is that in the current study, this assertion is assumed but not directly demonstrated.

Unlike the authors’ 2022 work, which included detailed behavioral validation (e.g., first-visit accuracy, anticipatory licking, trial-by-trial learning curves), the present manuscript provides no quantification of learning or memory use in the current dataset. This omission leaves open the possibility that rats may not have engaged spatial memory consistently.

In addition, the central replay analyses appear to pool across all trial types, without separating Home trials (which require memory) from Random trials (which are cue-guided). In my view, this makes it impossible to determine whether replay events, particularly ripple-less ones, occurred in memory-relevant contexts — undermining the core claim that ripple-less replay contributes to memory consolidation.

While the authors present detailed analyses of ripple field structure and place cell recruitment, these do not address the main concern. The original critique —that the study lacks a careful theoretical and experimental decomposition of the behavior— refers not to the spatial structure of the arena, but to the cognitive interpretation of replay events. If I correctly understood the task, only Home trials require memory, then it is essential to demonstrate that: rats were actively using spatial memory during these trials, and replay events (especially ripple-less ones) occurred during such memory-guided behavior. The authors should address the following main concerns:

1. It remains unclear whether the dataset used here is identical to that of Widloski & Foster (2022), a subset, or an expanded set. The manuscript should clarify:
 - a. Whether new data were included
 - b. Which sessions or animals were analyzed,
 - c. What criteria were used for inclusion or exclusion.
2. Include behavioral metrics from the current dataset to demonstrate that rats learned and used the Home well location. E.g., first-visit accuracy to the home well, trial-by-trial improvement in performance (e.g., reduced latency), anticipatory behavior (e.g., waiting or licking at the Home location before reward delivery).
3. Replay events should be separated by trial type (Home vs. Random) to establish which replays occurred in memory-guided contexts. This distinction is essential to support claims of mnemonic function. Explicitly analyze whether ripple-less replays occur preferentially during Home trials. This would directly support the claim that such replays relate to spatial memory or consolidation.

Version 2:

Reviewer comments:

Reviewer #1

(Remarks to the Author)

The authors have significantly improved what was already an exciting paper.

Reviewer #2

(Remarks to the Author)

Reviewer #3

(Remarks to the Author)

I recommend acceptance of the manuscript “Replay without sharp wave ripples in a spatial memory task.” The study offers rigorous and novel insights into hippocampal replay mechanisms and meets the journal’s standards for publication.

REVIEWER COMMENTS

Reviewer #1 (Remarks to the Author):

Summary

The title of this work refers to an observation of a somewhat-weakly supported phenomenon ("Replay without sharp wave ripples"). However, the subsequent analysis of the structure of population bursts / ripple oscillations during sequences of spiking produced when animals are paused during a continuous exploration task is very exciting and potentially of high significance. Taken together, this work undoubtedly merits being reported in this platform.

Sequences generated in the hippocampus have been conceptually identified as potentially critical features for learning and memory - facilitating long term storage, supporting recall, or even simulating potential future behaviors based on a model of the environment. However, a significant portion of the evidence for these ideas relies on a bit of slight-of-hand - the authors talk about sequential activation, but then focus on sharp wave ripples. Sharp wave ripples are a high-frequency oscillation associated with highly synchronous bursts of activity from ensembles of hippocampal neurons (population burst events, PBEs). Hippocampal replay during slow-wave sleep and awake quiescence has long been asserted to co-occur with SWR/PBEs. (SWR are defined based on local field potentials and PBEs are defined based on population firing.)

While previous authors have pointed out that not all SWR occur during a PBE and vice versa, and that hippocampal sequences can continue across chains of SWR, this work for the first time focuses on analyzing hippocampal activity during stopping periods based on decoded trajectories, and finds that many stopping periods are associated with continuous decoded trajectories which may or may not contain SWR. Moreover and very excitingly, they find that the SWR occur during decoded trajectories in consistent spatial locations associated with spatial learning. Again, while previous work had identified that SWR-linked replay contains a non-uniform distributions of represented spatial locations, this was an isolated observation that did not recognize that these ripples might be occurring as segments in part of longer events. Thus, this work will focus further investigations into the distinction between replay and SWR as complimentary but potentially functionally and mechanistically distinct phenomena.

While my enthusiasm for the work is high, the manuscript could be improved by clarify a number of methodological uncertainties.

We really appreciate the enthusiasm of the reviewer and the detailed examination of the content of the manuscript. We hope to address each of their concerns below.

Major Concerns:

(A) The overarching claim that 25% of replays contained neither ripples nor bursts rests on the definition of (1) replays, (2) ripples, and (3) bursts.

(A1) Replays are defined as periods when the Bayesian posterior (from place cells) meet 3 criteria - speed < 5 cm/s, posterior spread < 10 cm, posterior COM jumps < 20 cm. While the arena size is not given, previous work suggests it is ~90 cm on a side. Thus the example event in Figure 1A comprises a trajectory of 90 cm over 500 ms, or a speed of 180 cm/s (which is >> 5!). Presumably, this implies that the speed threshold is a typo of some sort.

Apologies for the confusion. "Speed" here refers to "rat speed". There is no requirement on the speed of replay.

(A1b) Replay segments are combined if the gap between them is < 20 cm and < 50 ms. But no minimum size of a segment is described. So this seems to suggest that as the 5 ms sliding window decodes slide along, two could connect 50 ms apart and create a replay segment even if the intervening 40 ms of bins did not satisfy the other criteria. Since the minimum duration of a replay segment is 100 ms, this suggests that an event could be constructed by three 80 ms windows, each 40 ms apart, which happened to have nearby decoded positions. ****Critically**** these special moments could largely be during ripples, and the smooth trajectories between them could just be the interpolation generated by sliding windows. It is unclear how to increase confidence here, but it would be very reassuring if non-overlapping bins generated consistent results?

We appreciate the concern here. To address this, we have re-decoded all replay events using non-overlapping 20 msec bins and have then recomputed the ripple fields. Below is one example session, with the original ripple field (below, left) and the recomputed ripple field (below, middle). As is apparent, ripple field locations remain largely intact. These fields are highly correlated across sessions (below, right), with all session correlation scores passing shuffle tests.

We now include these panels as Figure S4A-C, and address this point in the main text (204-207):

"We also ruled out trivial explanations for ripple fields: Ripple fields were not artifacts of our decoding approach -- using non-overlapping 20 msec time bins to re-compute replay trajectories yielded similar ripple fields (37 out of 37 sessions, $p < 0.001$; Figure S4B-C)."

(A1c) Unlike classic replay analysis, no effort appears to be made here to identify portions of decoded trajectories that are "significant". While that is sensible, the big concern is that non-SWR replay is being detected effectively by chance - spikes from a small number of neurons are able to drive the appearance of a sequence. Rather than asking for shuffle analyses, is it possible to justify NOT doing them? At minimum, it would be worth reporting how the replay criteria (speed, jump distance, posterior spread) compare with shuffled data.

We appreciate this important point and agree that a shuffle analysis is needed. All replays are now required to pass two shuffle tests, as follows: Each event is re-decoded 100 times using shuffled cell IDs, and the mean posterior spread and jump size is re-computed. A p-value for each measure is computed as the fraction of shuffles that have mean posterior spread or jump size less than the actual value. Each replay is required to have p-values for both measures less than 0.05. This filtering is now described in the Methods section as follows (lines 475-479):

"A candidate replay (merged or not) was denoted a *replay* if: (1) its duration was greater than 100 msec, and (2) it's spatial dispersion was greater than 12 cm, and (3) it passed a place cell-ID shuffle test: Each event was re-decoded using shuffled place-cell IDs 100 times, and the mean posterior spread and jump size was computed for each event. P-values for each replay were computed as the fraction of shuffle events less than the actual value for both measures. Replays were required to have p-values for each measure less than 0.05."

The distribution of p-values are plotted below. Filtering our original dataset for significance eliminated about 10% of replays.

Using this subset of replays, we found results similar to our original report, both in the fraction of ripple/burst-less replays (at ~21% – see below, center – compared to ~22%) and LFP properties (below, right, note the absence of ripple power in the blue curve and the similarity between the blue, green, and yellow curves). Below are the new panels for Figure 1H, N.

We also show that for significant replays, the mean posterior spread and jump size for ripple/burst-less replays is significantly less than for shuffles and comparable to the difference observed for replays w/ bursts or ripples.

These panels above now comprise Figure S2H-I, and are referenced in the main text (lines 187-189) as follows:

"Ripple/burst-less replays also showed similar low levels of spiking compared to post-replay control periods (Figure 1O, Figure S2G-H), this notwithstanding the fact that the quality of decoding during ripple/bursts-less replays vs. replays with bursts or ripples was comparable **and substantially improved over replay-length windows taken post-replay or at random (Figure 1P-Q), or the same replay windows decoded using shuffled place-cell IDs (Figure S2I-J).**"

Lastly, we asked how many ripple/burst-less replays would be expected by chance if we were to shuffle place-cell IDs and then look for new replay events (as with the original replays, these events were also required to pass the two shuffle tests for replay content). Below, left shows, for a single session, the number of ripple/burst-less replays across 100 place cell ID shuffles (solid line), compared to the number in the unshuffled data (vertical dashed line). Computing p-values as the fraction of shuffles with counts larger than the actual value, we found that most sessions (27 out of 37 sessions) had significant numbers of ripple/burst-less replays compared to chance. The combined p-value was less than 0.001, computed from a Fisher combined probability test.

We now include these panels as Figure S2A-B in the manuscript, along with the following line in the Results section (lines 170-174):

"Also, we computed the number of ripple/burst-less replays expected by chance after shuffling place-cell IDs and detecting new replay events (100 times). We found that across sessions, the number of ripple/burst-less replays in the original data was significantly larger than chance (21 out of 37 sessions; $p < 0.001$, Fisher combined probability test)"

(A2) Is the z-scoring of ripple power done only during periods when the animal is stopped, or the whole session? Does it also include sleep epochs not discussed in this paper? What about periods of noise from, e.g., scratching or chewing artifacts?

Z-scoring was performed based on the data from stopping periods, which were visually inspected (i.e., that the peak events looked reasonable) before normalization. Sleep sessions were not included in any of the analysis of this paper. We now address these concerns in the manuscript (lines 490 - 492):

"Peak events in stopping period data were visually inspected to make sure that z-scoring was not biased by "noise" events (e.g., chewing artifacts, implant collisions, scratching, etc.)."

(A3) Population burst events are described as drawing from "all clusters". Does this include multiunit activity? Clear artifact/noise? Obvious interneurons? Are the results roughly equivalent if PBEs are defined just based on pyramidal cells?

We apologize for the confusion on this point. By "cluster" we mean that the unit must pass three cluster quality metrics: noise overlap, isolation quality, and peak SNR. This definition would include interneurons but exclude multiunit activity and artifacts/noise. We now clarify this point in the Methods, lines 484-486:

"Population spike density was computed by first summing the total number of spikes from all clusters within a session (i.e., clusters with noise overlap < 0.03, isolation > 0.95, peak SNR > 1.5) in 1 msec non-overlapping time bins."

If we restrict burst events to drawing from place cells only by requiring that spatial information > 0.5 and mean firing rate > 0.05 spikes/sec, we find nearly identical results, i.e. that the fraction of ripple/burst-less replays is about the same (~22%):

We now mention this in passing in the main text (line 159):

"Incorporating this more sensitive criteria (burst detection, ripple detection on tetrode-averaged ripple power, and ripple detection on at least 11 tetrodes), we found that approximately 22 +-2% of replays contained neither ripples or bursts (Figure 1H; **we found similar results when bursts were computed using only place cells – not shown**)."

We also confirm that our findings with regards to "spike density fields" having significant spatial information and stability are also largely unchanged when we use only place cells to define spike density. Similarly, we mention this in passing in the main text (line 225):

"In contrast, spatial fields derived from the local population spiking on each tetrode were highly variable across tetrodes (Figure 3B,D), indicating a dissociation between the local LFP and local spiking. However, the spike density field derived from all clusters across tetrodes (Figure 3C) was strongly correlated with the ripple field (37 out of 37 sessions, $p < 0.001$; Figure 3E) and showed similar levels of significant spatial information and stability (36 out of 37 sessions, $p < 0.001$, and 36 out of 37 sessions, $p < 0.001$, significant; **we found similar results when spike density fields were computed using only place cells – not shown**)"

(A4) "... approximately 25% of replays contained neither ripples or bursts". It would be clearer to say that "~25% of replay segments shared no overlap with ripple or population burst event periods." ("Contained" might alternatively mean "completely included" rather than "overlapped with".)

Point taken. For concision we prefer the word "contain" but have added the following disclaimer in both the Methods (line 500) and main text (line 128):

"A given replay was considered ripple/burst-less if (1) it did not contain a ripple or burst event (**by "contain" we mean that the time of peak ripple power or bursting was not within the replay**) and (2) the peak ripple power and peak spike density were not greater than chance when compared to random snippets of data outside of replays taken throughout the stopping period (see Methods)"

(B) *The definition of a ripple occurrence using the average ripple power across all tetrodes is probably not consistent with the literature. The "bimodal" distribution in Figure 1E has a peak at 50 tetrodes, which is vastly more than most historic recording experiments. This needs to be acknowledged in the discussion. (It also is unclear that it is bimodal?)*
Moreover, the idea that there are "chance" ripples is a semantic sleight-of-hand. It would be much fairer just to structure this paragraph around the idea that many SWR occur as small events on a limited number of tetrodes, but that "large ripples" or "ensemble ripples" represent a unique class with special properties.

We adopted the approach of averaging ripple power across all tetrodes since it was used in Pfeiffer and Foster, 2013 (we now cite them). Averaging this way is admittedly a coarse approach, which is why we also searched for ripples independently across tetrodes, and

required that ripple-less replays pass criteria for ripple detection on both individual tetrodes as well as averaged across tetrodes (see lines 150-151).

We agree that the term "bimodal" is technically inappropriate here given the absence of two clear peaks in the data. We have replaced it with the term "U-shaped", which indicates that there is a clear separation between low-tetrode- and high-tetrode-participating ripple events. The reason we are not willing to classify low-tetrode, low amplitude events as a class of ripples unto themselves is because we cannot distinguish them from truly chance events like, e.g., locally filtered action potentials that masquerade as ripples (Liu et al., 2022). Assuming these events occurred independently across tetrodes, they would cluster at the left end of the distribution. Consistent with this hypothesis, we found a similar distribution when shifting our detection window to the relatively quiet periods post replay. While we cannot rule out the possibility of low-tetrode ripple events (i.e., those involving less than 11 tetrodes, a threshold we compute from the binomial distribution based on the ripple detection rate per tetrode across random snippets of time), for all intents and purposes we cannot measure them. To help clarify these points we have added the following to the manuscript (lines 143-149):

"We suspected that the low-amplitude, low-tetrode events could be chance fluctuations in ripple power due to, e.g., locally filtered action potentials (Liu et al., 2022), which, if occurring independently across tetrodes, would similarly cluster at the lower end of the distribution. Consistent with this hypothesis, repeating the analysis on windows immediately post-replay (+0.75 sec) where ripples are unlikely to occur (Figure S2A) yielded only low-tetrode ripple events (Figure 1E, green). Thus, while we cannot rule out the possibility of ripple events detected on very few tetrodes, for all intents and purposes we cannot distinguish them from chance. To establish a false positive baseline..."

Also, we also acknowledge our large number of tetrodes, and the high coincidence of ripple detection across them, in the discussion (lines 280):

"We developed a replay detection method that did not require detection of ripple or population activity burst and showed that replay can occur without ripples or bursts. To confirm the absence of ripples, we looked for ripples local to one or a few tetrodes, which we were well placed to see given our 64 tetrodes, a quantity that is vastly more than in previous work. While we found that many ripple events during replay were detected by large numbers of tetrodes, ripples that were detected by small numbers of tetrodes were no more likely than during an equivalent period of time just after each replay, when ripples are unlikely to occur – i.e., these detections are below the noise threshold."

(C) I could not understand Figure 1H (are the lighter salmon colors bars that are overlapping the darker "both" bar or separate, taller bars?), and the discussion was also quite confusing. I think the upshot is that SWR and PBEs largely measure the same phenomenon. This is not surprising, but perhaps since the Malvache, 2016 reference is in the literature, it is worth including. (Also relevant for cross-hemisphere correlation, which you could report as a useful

number!) But I think that reporting it as unexpected or unlikely makes it hard on a reader who is used to the two being considered the same events.

The lighter salmon colors count non-overlapping event categories within the larger category of replays that have either a ripple or a burst, the vast majority of which contain both. We have modified the panel to clarify this:

We also agree that SWRs and PBEs probably measure the same phenomenon, but the co-incidence rates in previous reports tend to be underwhelming: Pfeiffer & Foster report around 50% (supp. Figure 8), O'Neill et al. (2017) report around 29.9% (Fig 3H), and Malvache et al. (2016) around 25-45%. We now cite these additional references in the introduction of the main text (lines 78-79):

"While not every ripple-based replay necessarily contains a burst, and not every burst-based replay contains a ripple (the probability of coincidence tends to be moderate: e.g., 50% in Pfeiffer & Foster, 2013; 25-45% from Malvache et al., 2016; 29% in O'Neill et al., 2017), nonetheless these methods identify candidate replay events based on the detection of one or the other"

We also indicate that the high coincidence rate we measure is not unexpected (lines 168):

"In contrast, we found that ~80% of replays that contained either a ripple or a burst contained both (Figure 1H), indicating that ripple and burst events are highly coincident as expected."

(D) It is important to report the way that LFP signals are electrically referenced, and how electrode positions are inferred particularly when discussing averaging/comparing ripple signals across different tetrodes. Is it valid to average ripple band power values from tetrodes in s. pyramidae and s. radiatum, for instance???

We have now added referencing information to the Methods (line 402):

"Spikes were extracted from channel LFP's, sampled at 30 kHz and referenced against a tetrode placed in the corpus collosum (one for each hemisphere), using Spike Gadgets Trodes software..."

Ripples in s. radiatum are substantially reduced in amplitude compared to s. pyramidae (Liu et al., 2022), so it is unlikely we are detecting any significant ripple events there. That said, the radiatum may reflect ripples generated in CA3 that may be independent from those generated locally within CA1. For our purposes, this should broaden the sensitivity of our ripple detection approach (with individual tetrodes or averaged across tetrodes).

(E) How do ripple fields correspond with spatial occupancy?

We have already shown that ripple fields do not correlate with the rat's position density during replay (Figure 1J). However, there is a moderate correlation with the rat's position density during movement (15 out of 37 sessions significant; below).

However, closer inspection of the movement density across sessions shows that it doesn't really capture the idiosyncratic structure of the ripple fields (below, 1st and 2nd columns), whereas the similarity between place field sums and ripple fields is much more compelling (1st and 3rd columns).

These panels now comprise Figures S4M,R in the manuscript. We also address this in the Results section (line 244-251):

"In contrast, ripple fields were moderately correlated with the rat's trajectory density during movement (15 out of 37 sessions, $p < 0.001$; Figure S4M,R), indicating that ripple fields tended to occupy the corridors of movement through the arena. However, visual inspection of the movement trajectory densities across sessions showed that they often failed to account for the idiosyncratic structure of ripple fields (compare Figure S4H to S4M) that was clearly preserved across rats experiencing the same barrier configuration (Figure 4A), begging the question of whether a stronger correlate of ripple field locations could be found"

Minor Concerns:

(1) *The arena size is not specified in the text or in the figures (i.e., as a scale).*

We have added the arena size to the main text (line 108).

(2) The length of the filter for the Gaussian smoothing kernels should be reported, not just the SD.

We have added the filter size to the description of each smoothing kernel in the text.

(3) Can the rats hear the next reward well being filled? Does this occur during exploration or when they are still? Is it associated with the location of the ripple fields? What about their average location when the reward light is illuminated?

We believe the rats cannot hear the reward well being filled for several reasons. First, the solenoid valves used to dispense reward were essentially noiseless (one has to have their ear against it to hear any sound). Second, all 9 valves were grouped together in a bank several feet beneath the floor of the elevated maze. Third, there was a loud air vent that operated continuously in the recording room, which provided a strong mask to other sound sources in the room.

Rewards were dispensed after a variable time delay (5-15 seconds) after the rat had left the previous well. The reward lights were only illuminated on Random trials, and coincided with the filling of the random well. Because we never observed any behavioral evidence that the rats paid any attention to the lights and the fact that acquiring these time points would require a serious effort (due to oversight, they were not saved and would have to be extracted from the video), we have chosen not to pursue this analysis.

(4) The discussion should probably discuss "ripples that aren't replay", and how they fit into the story. For example events when the posterior is still or loops back on itself.

Thank you for pointing this out. To address them, we have added the following statement to the Discussion section (lines 300-302):

"Note that our replay detection criteria exclude stationary replays (i.e., non-moving reactivations; Denovellis et al., 2021), which may relate to ripples and bursts according to different organizing principles than the events we have considered here."

(5) "We restricted analysis to all replays associated with reward consumption" - this could be more clearly phrased based on a definition.

We have included a description of how reward consumption times were defined in the Methods (lines 395-399):

"Periods of reward consumption were defined as times in which the smoothed rat speed (a second order Butterworth filter with a cutoff frequency of 0.02 samples/s applied to the rat's speed computed above, i.e., it was smoothed a second time), dropped below 1 cm/s while the rat was at the rewarded well."

(6) "To establish a false positive baseline, we performed the same ripple-detection analysis on post-replay windows (0.75 sec after the end of the replay), where we determined ripples to occur at a minimum (Figure S2A)." Both black and green lines speak of special neural states. It is a bit confusing using the post-replay window as the false positive baseline, especially when this window is selected by finding the minimum ripple detection. Why not randomly select windows of the same length of replays (like Figure S2B)?

We misspoke here. The false positive baseline used to set the minimum tetrode count was actually derived from the analysis presented in Figure S2B using randomly selected replay-length windows from the stopping period. The post-replay ripple detection analysis was performed to understand the source of the low-amplitude, low-tetrode ripple events that we measured in Figure 1E (see response to question (B) above). We have modified the text to clarify that the baseline comes from the shuffle analysis (line 149):

"To establish a false positive baseline, we first computed the probability of ripple detection per tetrode during stopping periods – for each tetrode, and for each replay, 100 replay-length windows were taken randomly across stopping periods and evaluated for ripples – which we found to be ~12%. The 95th percentile of the binomial distribution with this success probability across 64 detectors (Figure S2B) occurred at 11 tetrodes, indicating that to detect a ripple event confidently, at least 11 tetrodes must detect it."

On a related note, we have also taken the opportunity to include detailed analysis of randomly selected replay windows during stopping periods, shown now in yellow below (Figure 1N-Q), and strengthen our argument that ripple/burst-less replays lack the distinctive signs typical of most replays with respect to LFP or spike density properties (N-O), they nonetheless carry significant spatial content (P-Q).

(7) Page 13, "An interesting consequence of our finding is that, to the extent that ripple expression is gated by individual spatial fields, the patterns of neural activity driven by ripples may likewise resemble relatively static representations rather than sequential trajectories." What is the corresponding result that supports this statement?

We apologize for the ambiguity of this line. We simply mean that ripples do not circumscribe full replayed trajectories (they are not one-to-one) but rather occur within spatially restricted and static portions of the replayed environment. This is supported by the fact that ripple fields have significant spatial information and stability. To clarify this, we have edited that line as follows (line 327):

"An interesting consequence of our finding is that, to the extent that ripple expression is gated by individual spatial fields, the patterns of neural activity driven by ripples may likewise resemble relatively static representations rather than **full replayed** trajectories."

(8) (Methods) What is the unit for spatial information? It is said to be in bit/spike but the SI criteria for identifying a place cell is in bit/second.

Thank you for catching this. The units should be bits/spike. We have added the units to the figures and corrected them in the Methods.

(9) Replay detection without SWR/PBE event detection first has been used before, e.g., Carey, Tanaka, van der Meer 2019 has a "sequence detector". Should be cited/mentioned?

While indeed candidate sequences are constructed in the same way (by decoding the full session and looking for spatially continuous portions of the decoded locations), ultimately a ripple overlap criteria is applied, as quoted here:

"The resulting pool of candidate sequences is then subjected to further criteria, starting with a minimum length (50 ms), a minimum number of neurons participating in the event (4) and overlap with a putative SWR for any nonzero amount of time (see later in the article for how these are detected)".

Thus, we have chosen to cite them when referencing existing replay detection methods based on ripples (line 76).

(10) Figure 1G needs a unit (seconds?).

Corrected.

(11) Figure 5 B/C captions don't match labels in figure.

Corrected

Reviewer #2 (Remarks to the Author):

Reviewer #3 (Remarks to the Author):

Widloski and Foster's study provides evidence for the existence of ripple-less replays, which they describe as a functionally distinct subset of hippocampal activity. These replays were characterized by smooth, continuous spatial trajectories yet lacked the sharp-wave ripples traditionally associated with hippocampal memory processes. While their temporal properties suggest a potential role in hippocampal function, the claim of functional distinctiveness is not directly tested, leaving room for alternative interpretations such as their alignment with other hippocampal phenomena like theta sequences.

We agree that it is very easy to mistake theta sequences for ripple/burst-less replays. We have taken several steps to ensure that this is not the case. First, as we showed in the original manuscript, theta power is much *smaller* during ripple/burst-less replays than replays with burst or ripples (compare blue vs red, below; these are Figure 1M and Figure S2C), but comparable to other "quiet" times during the immobility period outside of replay (green), in contrast to what would be expected during theta sequences.

We have also performed two new analyses. We reasoned that if ripple/burst-less replays are just theta sequences, then compared to replays w/ ripples or bursts, they should (1) move slower (compare Gupta et al., 2012 to Davidson et al., 2010) and (2) orient more towards the rat's future path. In contrast, we found that replay speeds were comparable (below, left), and that the angular displacement to the rat's future behavior was larger for ripple/burst-less replays (below, right).

Thus we believe we can confidently rule out that these events are theta sequences. We have included the panel at left as Figure 1L.

Furthermore, the concept of "ripple fields," spatially stable zones where ripple-associated replays occur, is introduced as a mechanism for amplifying salient experiences during memory consolidation. However, the connection between ripple fields and learning or memory remains hypothetical, as no direct link to improved task performance or learning outcomes is provided. Without metrics such as accuracy or learning curves, it is unclear whether the three rats genuinely learned the task or relied on simpler, non-mnemonic strategies.

We had quantified learning across trials in Figure 1D-E of a previous publication (Widloski & Foster, 2022). Rats learned to identify and locate the Home well during the "Home trials" as evidenced by the fact that the probability of immediately visiting the Home well was higher than for Random wells (panel D). Rats also understood the alternating structure of the task, as evidenced by the fact that only on Home trials does the rat go to the Home well and wait for the reward (the reward is dispensed with a 5-15 sec delay after the start of the trial). All of this learning occurs rapidly, reaching significance after about 5 trials.

To address this in the manuscript, we now state explicitly a reference to this work with respect to behavioral learning on the task (lines 221- 223):

"In our previous work (Widloski & Foster, 2022), we showed that rats learn the location of the Home well after only a few trials, navigating to it quickly even as the barriers changed unpredictably across sessions. This behavior suggests that the rats internalize knowledge of the environment to move efficiently through the environment (supported also by the fact that replays conform to the barriers in each session – Widloski & Foster, 2022)."

Furthermore, the study lacks a careful theoretical and experimental decomposition of the behavior, which is crucial for understanding how replay without sharp-wave ripples relates to functional outcomes.

A central finding of our paper is that, with respect to ripple-less replays, it is more parsimonious to think about ripple fields – whether or not a ripple accompanies a replay depends on the replays' proximity to the ripple field as it moves through the environment. To remind the reviewer, we have already performed an exhaustive but ultimately fruitless search for strong behavioral correlates of ripple fields, including:

1. Proximity to salient locations in the environment, like the reward wells and barrier locations (Figure S5A-B)

2. Locations where the behavioral affordances of the environment, as measured by the barrier locations or from the behavior itself, have changed the most (Figure S5C-F)
3. Rat trajectory density during movement and immobility (Figure 1J, 1R)

Thus, we think that it is unfair to suggest that a careful decomposition of the behavior has not been performed. Later, we show that ripple field locations are most strongly tied to where place cells are most represented, in particular the unstable ones (Figure 5). This is consistent with previous reports that tie ripples to hippocampal map maintenance and updating, which we discuss in the Discussion section.

REVIEWER COMMENTS

Reviewer #1 (Remarks to the Author):

We remain very enthusiastic about this paper. The remarkable data set that the authors have gathered permits them to study sequential non-theta patterns in hippocampus in a way that provides much better grounding than we have had previously. In addition, the "ripple field" phenomena seems potentially important and will merit further study by the field. Most importantly, clearly articulating the independence of sharp wave ripples and sequence generation will be of critical relevance to our understanding of how the system works for memory consolidation and integration.

That said, we did have some residual concerns in the most recent draft we have received.

(1) In the textual introduction to replay detection, "114-115 decode the posterior probability of position from the spiking of all simultaneously recorded cells across time bins of size 80 msec throughout the session. During stopping periods in the task, candidate events were identified as continuous epochs lasting at least 100 msec" - This (80 vs 100) will confuse the reader and force them to discover the fact that a sliding window is employed. Consider adding this information here?

Thanks for the suggestion. We have modified the text to indicate that the decoding windows are overlapping (lines 112-115):

"Place fields were determined for each active cell by normalizing the spatial distribution of rat positions at spike times during run by the rat position occupancy, and memory-less Bayesian position estimation was used to decode the posterior probability of position from the spiking of all simultaneously recorded cells across **overlapping** time bins of size 80 msec (**shifted by 5 msec**) throughout the session."

(2) Now that replay detection is clarified to use a shuffle test, Figure 1P,Q /S2I,J and the text callouts (lines 185-190) are problematic. First, the replay events are selected by shuffle testing the posterior spread and jump distance to be small. It thus seems unsurprising (and almost a tautology) that they would be smaller than randomly chosen data (green and yellow) or particularly compared to shuffled data (S2I,J)? Second, in particular, the claim "the quality of decoding during ripple/bursts-less replays vs. replays with bursts or ripples was comparable" seems unsupported by the data that shows a highly statistically significant difference between them. This claim is paired against the first claim ("they are also much smaller than random"), but as pointed out, that's obvious by construction. Perhaps the idea that you hope to demonstrate to the reader - that these are also real replay - could be made by making a statement about how much the distributions overlap? ("the median ripple/burtless replay had a quality greater than 40% of replays with bursts or ripples").

Thank you for noticing this. The comparison to shuffled data is indeed a tautology, since we used place field shuffles to define our replay events. Thus, we have chosen to remove these panels (and references to them) from the manuscript.

We agree it is unsurprising that decoding quality for replay is significantly improved over windows taken at random or after a delay (green and yellow), but we have opted to keep it in as it illustrates that the peak spiking within a window gives little indication as to whether it contains a replay content or not. We have amended the text (lines 189-193) as follows:

"Ripple/burst-less replays also showed similar low levels of peak spiking compared to post-replay control periods (Figure 1O, Figure S3F-G), this notwithstanding the fact that the quality of decoding during ripple/bursts-less replays ~~vs. replays with bursts or ripples~~ was ~~comparable and~~ substantially improved over replay-length windows taken post-replay or at random (Figure 1P-Q). **This indicates that peak spiking alone provides little information about whether a window contains a replay event.**"

We also apologize for overstating the similarity in decoding quality between replays with and without bursts or ripples. Because ripple-less replays naturally involve less spiking, they might appear to have poorer decoding quality simply due to lower spike counts. To control for this, we matched the two groups on coarse spiking properties—specifically, spikes per second and number of active cells per second across replays (see below). We did this by constructing 2D histograms over these two measures for each group and then downsampling events so that the distributions were identical.

We found that this equalized the decoding quality across the two replay types ...

... while crucially preserving differences in peak ripple power and spike density and theta power (below), indicating that the presence of ripples or population bursts is not required for the generation of high-quality, temporally structured replay events.

These panels now comprise **Figure S3H-N**. We have modified the main text as follows (lines 193-202):

"We initially observed a significant difference in decoding quality between replays with and without bursts or ripples (Figure 1P-Q). To account for the possibility that this difference was driven by lower spike rates in ripple/burst-less events, we matched the two replay types on coarse spiking properties—specifically, spikes per second and

number of active cells per second (Figure S3F-G). This was done by constructing a two-dimensional histogram over both measures for each group and downsampling events so that the resulting histograms were identical. This matching procedure equalized decoding quality across the two replay types (Figure S3H-I), while preserving key differences in ripple power, spike density, and theta power (Figure S3J-L), suggesting that the presence of ripples or population bursts is not required for the generation of high-quality, temporally structured replay events.

We also remind the reviewer, in case it was missed in the last round, that we also did a separate analysis showing that for most sessions (27 out of 37 sessions), the number of ripple/burst-less replays we detected was significantly more than the numbers we observe after place fields are shuffled (Figure S3C-D). We hope that together these results are sufficient to demonstrate our claim that ripple/burst-less replays are a real phenomenon.

Reviewer #2 (Remarks to the Author):

Reviewer #3 (Remarks to the Author):

Here, I restate my core concerns in a more comprehensive and focused manner. I hope this clarifies my original intent, and I apologize if these concerns were not expressed clearly enough in the initial round of review.

The main claim of the manuscript is that hippocampal replay can occur in the absence of sharp wave ripples and that ripple occurrence is spatially organized into stable “ripple fields” in decoded space. The authors propose that ripples selectively tag behaviorally relevant replays, particularly those associated with learning or novelty, challenging the conventional view that replay and ripples are functionally inseparable.

The manuscript presents compelling electrophysiological evidence in support of this novel framework. However, the interpretation that ripple-less replays reflect memory-related processes hinges critically on the assumption that animals were engaged in a spatial memory task. The problem is that in the current study, this assertion is assumed but not directly demonstrated. Unlike the authors’ 2022 work, which included detailed behavioral validation (e.g., first-visit accuracy, anticipatory licking, trial-by-trial learning curves), the present manuscript provides no quantification of learning or memory use in the current dataset. This omission leaves open the possibility that rats may not have engaged spatial memory consistently.

In addition, the central replay analyses appear to pool across all trial types, without separating Home trials (which require memory) from Random trials (which are cue-guided). In my view, this makes it impossible to determine whether replay events, particularly ripple-less ones, occurred

in memory-relevant contexts — undermining the core claim that ripple-less replay contributes to memory consolidation.

While the authors present detailed analyses of ripple field structure and place cell recruitment, these do not address the main concern. The original critique —that the study lacks a careful theoretical and experimental decomposition of the behavior— refers not to the spatial structure of the arena, but to the cognitive interpretation of replay events. If I correctly understood the task, only Home trials require memory, then it is essential to demonstrate that: rats were actively using spatial memory during these trials, and replay events (especially ripple-less ones) occurred during such memory-guided behavior.

The authors should address the following main concerns:

1. It remains unclear whether the dataset used here is identical to that of Widloski & Foster (2022), a subset, or an expanded set. The manuscript should clarify:

- a. Whether new data were included*
- b. Which sessions or animals were analyzed,*
- c. What criteria were used for inclusion or exclusion.*

2. Include behavioral metrics from the current dataset to demonstrate that rats learned and used the Home well location. E.g., first-visit accuracy to the home well, trial-by-trial improvement in performance (e.g., reduced latency), anticipatory behavior (e.g., waiting or licking at the Home location before reward delivery).

Thank you for this clarification. In our 2022 paper, we had 4 animals, but as we describe in that paper, only 3 had enough place cells to do replay analysis. In this paper, we are using all sessions from the same 3 animals. We now clarify this in the first paragraph of the results section (lines 107-109):

"To determine whether replay content can exist in the absence of bursts or ripples, we utilized a previously published data set (Widloski & Foster, 2022) comprising of 6,580 replays recorded across 37 session from 3 rats and applied a novel replay detection method that did not depend on the occurrence of population bursts or ripples. **While the original dataset included 4 rats, only 3 had sufficient numbers of active place cells to support replay analysis; all sessions from these 3 animals were included in our analysis.**"

Further, we now show new behavioral analysis, namely in a trial-by-trial analysis of first visit accuracy and anticipatory licking at the Home well (see below) that supports our claim that these 3 rats performed the spatial memory task.

We have added the following to the main text (lines 108-113):

"Briefly, during each session, rats were trained on a spatial memory task to search for liquid chocolate in a square arena (90 x 90 cm) with movable barriers. Chocolate was available in one of 9 food wells, which alternated on consecutive trials between a learnable fixed location (Home well) and other unpredictable locations (Random wells). **Rats learned the location of the Home well after only a few trials, navigating to it quickly even as the barriers changed unpredictably across sessions (Figure S1).**"

We have also amended the Methods to explain how these behavioral measures are computed (lines 404-426).

3. Replay events should be separated by trial type (Home vs. Random) to establish which replays occurred in memory-guided contexts. This distinction is essential to support claims of mnemonic function. Explicitly analyze whether ripple-less replays occur preferentially during Home trials. This would directly support the claim that such replays relate to spatial memory or consolidation.

Thank you for this suggestion. We separated ripple/burst-less replay rates by trial type and found that at the Random well, before the rat is to make the return journey to the Home well, ripple/burst-less replay rates are in fact *boosted* (below, compare blue violin plots). At the same time, replays w/ ripples or bursts were slightly diminished at these locations, though the difference was insignificant (red). This is consistent with the notion that ripple-less replays play an important mnemonic function during goal-directed navigation. We now include this panel as **Figure S3J**.

We include the following text in the Results section (lines 202-205):

"Interestingly, ripple/burst-less replays tended to occur more often at a Random well compared to a Home well just prior to the goal-directed portion of the task (Figure S3J), indicating that these events are more prevalent during the memory-intensive portions of the task. Thus, our method allows us to reveal for the first time a large number of spatially extended, high quality, behaviorally significant replay events that would have gone undetected if classical methods of replay detection had been used."

We also mention it in the Discussion section (lines 332-334):

"Notably, ripple/burst-less replays were more prevalent during memory-intensive phases of the task, further emphasizing their potential functional significance."